# In vivo inducible reverse genetics in patients' tumors to identify individual therapeutic targets

Michela Carlet[1,15], Kerstin Völse[1,15], Jenny Vergalli[1], Martin Becker[1], Tobias Herold [1,2,3], Anja Arner[4], Daniela Senft[1], Vindi Jurinovic[1,2,4], Wen-Hsin Liu[1], Yuqiao Gao[1], Veronika Dill[5], Boris Fehse [6], Claudia D. Baldus[7], Lorenz Bastian [7], Lennart Lenk[8], Denis M. Schewe [8], Johannes W. Bagnoli[9], Binje Vick [1,3], Jan Philipp Schmid[1], Alexander Wilhelm[10], Rolf Marschalek [10], Philipp J. Jost[3,5,11], Cornelius Miething [12,13], Kristoffer Riecken [6], Marc Schmidt-Supprian[14], Vera Binder[4] & Irmela Jeremias [1,3,4 ✉]

High-throughput sequencing describes multiple alterations in individual tumors, but their functional relevance is often unclear. Clinic-close, individualized molecular model systems are required for functional validation and to identify therapeutic targets of high significance for each patient. Here, we establish a Cre-ER^T2-loxP (causes recombination, estrogen receptor mutant T2, locus of X-over P1) based inducible RNAi- (ribonucleic acid interference) mediated gene silencing system in patient-derived xenograft (PDX) models of acute leukemias in vivo. Mimicking anti-cancer therapy in patients, gene inhibition is initiated in mice harboring orthotopic tumors. In fluorochrome guided, competitive in vivo trials, silencing of the apoptosis regulator MCL1 (myeloid cell leukemia sequence 1) correlates to pharmacological MCL1 inhibition in patients´ tumors, demonstrating the ability of the method to detect therapeutic vulnerabilities. The technique identifies a major tumor-maintaining potency of the MLL-AF4 (mixed lineage leukemia, ALL1-fused gene from chromosome 4) fusion, restricted to samples carrying the translocation. DUX4 (double homeobox 4) plays an essential role in patients' leukemias carrying the recently described DUX4-IGH (immunoglobulin heavy chain) translocation, while the downstream mediator DDIT4L (DNA-damage-inducible transcript 4 like) is identified as therapeutic vulnerability. By individualizing functional genomics in established tumors in vivo, our technique decisively complements the value chain of precision oncology. Being broadly applicable to tumors of all kinds, it will considerably reinforce personalizing anti-cancer treatment in the future.

A full list of author affiliations appears at the end of the paper.

Translating comprehensive cancer sequencing results into targeted therapies has been limited by shortcomings of model systems and techniques for preclinical target validation[1,2]. The methodological gap contributes to the fact that only below 10% of drugs, successful in preclinical studies, pass early clinical evaluation and receive approval[3,4].

Functional genomic tools including RNA interference (RNAi) proved of utmost importance to annotate the numerous alterations detected by multi-omics profiling and significantly deepened our understanding of the merit of individual genes as drug targets[5,6]. As limitation, functional studies have largely been restricted to cancer cell lines, which often fall short in predicting the role of alterations in individual human tumors[7]. To approximate the situation of the patient, the predictive power of primary tumor cell cultures[8] and organoids[9] is currently under intense investigation[10].

For mirroring the clinical situation even closer, patient-derived xenograft (PDX) mouse models have been demonstrated to faithfully recapitulate the complexity of tumors in humans. PDX models are available for the vast majority of human cancers, and their preclinical value for biomarker identification and drug testing is well established[11–15]. It is increasingly recognized that the drug development process might profit from studying PDX models with molecular techniques, routinely used in cell line models and genetically engineered mouse models (GEMM)[16,17]. Still, RNAi techniques were only rarely applied for in vivo mechanistic studies in PDX, mainly due to technical challenges such as low transduction efficiencies and the need for continuous in vivo growth and associated high demand on resources[16]. As an advantage over constitutive systems, inducible gene silencing prevents overestimating in vivo gene function by avoiding influences from, e.g., transplantation and engraftment, and allows mimicking the treatment situation in patients with established tumors. The use of Cre-ER[T2]-loxP combines the properties of high ligand sensitivity while maintaining tight control of shRNA expression in the un-induced state, thus minimizing leakiness, an advantage over tet-regulated systems[16,18–20].

Here, we report a Cre-ER[T2] inducible RNAi in PDX models in vivo, using acute leukemia (AL) as prototype disease where ex vivo investigation on primary cells is challenging, but orthotopic PDX models are promising[21,22]. In proof of principle studies, we demonstrated that *MCL1* silencing in acute lymphoblastic leukemia (ALL) and acute myeloid leukemia (AML) PDX models correlates to response to pharmacological *MCL1* inhibition. We confirmed a tumor-maintaining potency of the *MLL-AF4* fusion protein in PDX models in vivo and used the technique to identify *DDIT4L* as therapeutic targets in PDX ALL carrying the recently described *DUX4-IGH* translocation.

## Results

### Development of a Cre-ER[T2] inducible shRNA knockdown approach in vivo.
To test the suitability of the inducible knockdown system across a broad range of leukemia subtypes, primary tumor cells from 5 patients with AL (3 pediatric ALL, 1 adult ALL, 1 adult AML; clinical patient data in Table S1) were transplanted into NOD scid gamma (NSG) mice (Fig. 1a). Resulting PDX cells were genetically engineered first with a construct encoding a Tamoxifen (TAM)-inducible variant of Cre-recombinase, Cre-ER[T2], together with a red fluorochrome for enriching transgenic cells and Gaussia luciferase (Luc) for bioluminescence in vivo imaging[23] (Fig. 1a). Transduction efficiencies were typically well below 30% (Table S2), putatively indicating a single viral integration per genome according to literature[24], leading to homogenous expression levels of Cre-ER[T2] (Fig. S1a), minimal toxicity and neglectable leakiness in all

samples, thus overcoming one of the challenges of TRE-based inducible expression systems[16].

In a second step, PDX cells were transduced with the small hairpin (sh) RNA expression vectors (Figs. 1a and S1b). The miR30-based knockdown cassette was directly coupled to a fluorochrome and both were cloned in antisense orientation, flanked by two pairs of loxP sites. In the absence of TAM, neither the inducible fluorochrome nor the shRNA were expressed. TAM administration induced a two-step Cre-ER[T2]-mediated recombination process which flipped the fluorochrome-shRNA insert into sense orientation, initiating its expression (Fig. S1b–c)[25,26]. A set of 4 recombinant fluorochromes was used to monitor shRNA transduction and recombination and to enable competitive in vivo assays (Fig. S1c). Transduction efficiency was tracked by iRFP the control vector encoding an shRNA targeting Renilla luciferase (shCTRL), or by mTagBFP in the vector encoding a gene of interest (GOI)-specific shRNA (shGOI) (Table S2). Upon TAM administration, Cre-ER[T2]-mediated recombination deleted the constitutively expressed fluorochromes iRFP and mTagBFP and induced expression of the second set of fluorochromes[27] (Figs. 1a and S1b–c). T-Sapphire and eGFP were chosen as inducible fluorochromes due to their high similarities in sequence and expression kinetics[28] and replaced iRFP and mTagBFP expression upon TAM treatment. The two knockdown vectors enabled pairwise competitive in vivo experiments in the same animal to increase reliability and sensitivity, while saving resources.

Mice were transplanted with a 1:1 mixture of PDX cells from the same patient expressing either of the two RNAi vectors, shCTRL or shGOI (Fig. 1a). For exemplary purposes and to describe distinct aspects of the method, the apoptosis regulator *MCL1* was chosen as GOI (Figs. 1 and S1). As quality control, expression of constitutive markers revealed equal engraftment of both populations at the time of TAM administration (Fig. 1b).

To induce gene silencing, TAM was administered to mice with pre-established leukemias when homing and initial engraftment to the murine bone marrow was achieved and PDX cells were in the exponential growth phase, mimicking treatment of patients with pre-existing tumors. Systemic TAM administration induced expression of the inducible fluorochromes T-Sapphire or eGFP, in similar amounts for both constructs, starting as early as 24 h, with highest expression levels obtained at 72 h after TAM (Fig. 1b). The functional consequences of control and GOI knockdown were monitored by quantifying each population according to their fluorochromes, using flow cytometry (Fig. 1b–c). TAM was dosed to obtain substantial Cre-ER[T2] induced recombination in the absence of toxicity and with recombination efficiencies independent of tumor load (Fig. S1d).

Several quality controls were performed to exclude unspecific toxicities; the distribution of both populations remained stable over time after TAM treatment, if both populations expressed shCTRL (shCTRL/shCTRL mixture in Fig. 1c, upper lane) in all PDX samples analyzed (Fig. S1d). Similarly, the distribution of the shCTRL/shGOI mixture remained unchanged, if mice received the carrier solution alone (Fig. S1f–g). These results are in line with our previous studies[29], where we found that transduction and enrichment of PDX cells was not associated with clonal selection, and that PDX samples largely maintained their sample-specific mutational pattern.

In contrast and upon treatment with TAM, the population expressing a shRNA targeting an essential GOI (sh*MCL1*) decreased over time and was overgrown by control cells (Fig. 1c, lower lane and Fig. S1h). Loss of cells with GOI knockdown in vivo proved a functional importance of the GOI on the molecular level, mimicking elimination of tumor cells in patients upon treatment with a targeted drug.

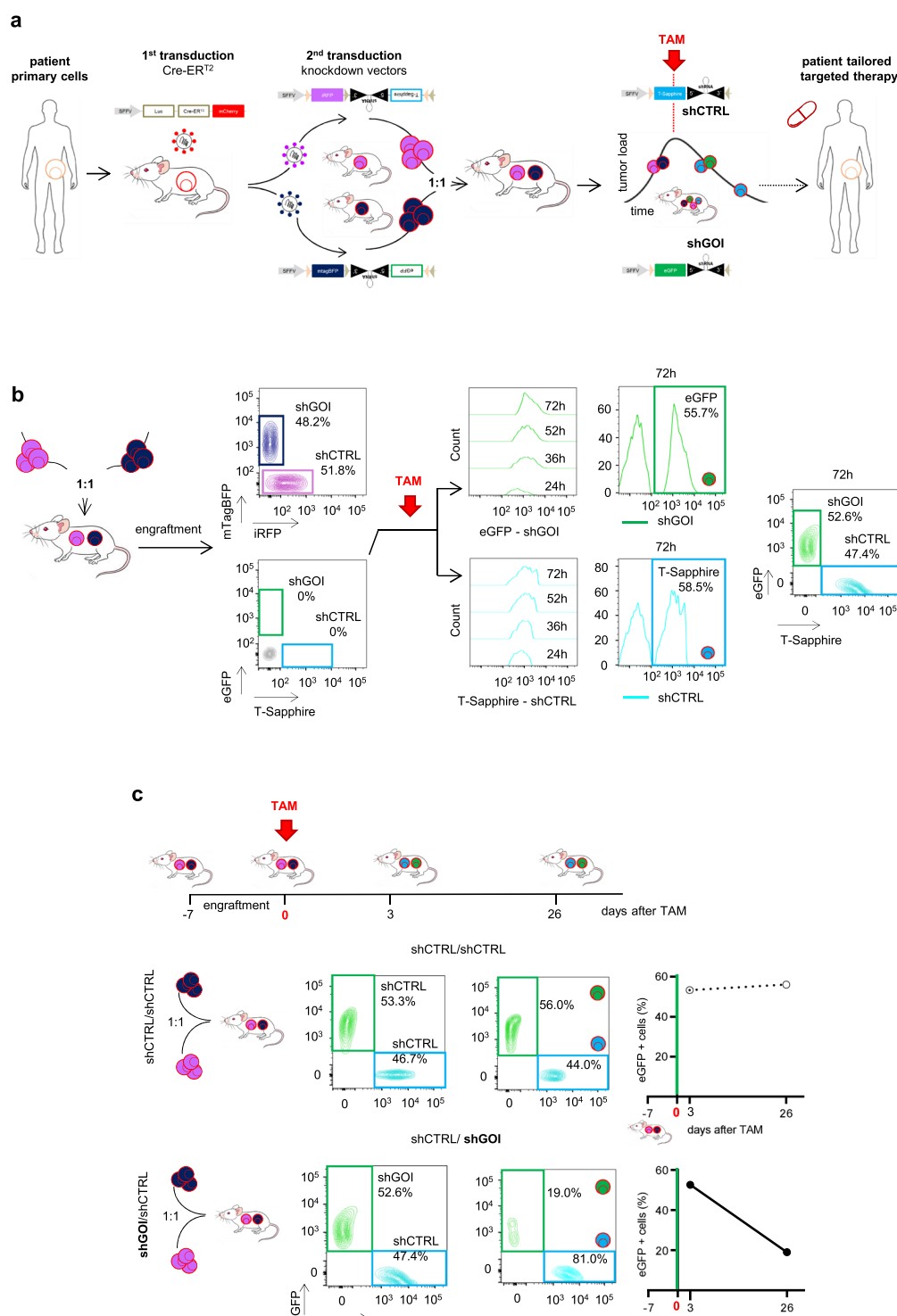

**Inducible silencing of *MCL1* correlates response to small molecule *MCL1* inhibitors in vivo**. To test whether inducible knockdown of the GOI correlates to targeted inhibitors, we first analyzed the response of PDX samples to shRNA-mediated inhibition of *MCL1*. We selected *MCL1* as proof of principle target gene from literature as certain, but not all leukemias seem responsive to *MCL1* inhibition[30,31]. The anti-apoptotic gene *MCL1* was chosen as it is dysregulated in numerous tumor entities[32] and *MCL1* inhibitors are currently investigated in clinical trials yielding mixed results[33] (NCT03218683). Predicting treatment response for selecting patients who will profit from

*MCL1* directed therapy remains a major challenge and functional in vivo assays might provide helpful insights[34].

We studied PDX models from three different patients with acute leukemia (AML-388, ALL-199, ALL-265). In the AML-388 PDX model, we found a clear decrease of cells with *MCL1* knockdown compared to control cells in vivo, accompanied by efficient knockdown on protein level (Fig. 2a–b), validating *MCL1* as important vulnerability. Importantly, these effects were independent of tumor load at the time of TAM administration, supporting the use of the inducible knockdown system at any disease stage (Fig. S2a). In contrast, knockdown of *MCL1* in two

**Fig. 1 Establishing an inducible knockdown system in PDX acute leukemia cells in vivo. a** Overview of the experimental setup: Primary acute leukemia (AL) cells were amplified in NSG mice and serially passaged PDX cells lentivirally transduced twice in a row; first to constitutively express Cre-ER[T2] together with mCherry and a luciferase (Luc); second to express inducible knockdown vectors containing (i) a constitutively expressed fluorochrome marker (either iRFP or mTagBFP) and (ii), placed in antisense orientation, a miR30-based knockdown cassette coupled to a second inducible fluorochrome (either T-Sapphire or eGFP). After amplification in mice, purified transgenic PDX cells were mixed 1:1 and transplanted into next recipient mice for competitive in vivo experiments. In mice with established leukemias, TAM was administered to induce Cre-ER[T2]-mediated recombination. Recombination inverted the knockdown cassette and induced (i) expression of the shRNA; (ii) deletion of the constitutive fluorochrome (either iRFP or mTagBFP) and (iii) expression of the inducible fluorochrome (either T-Sapphire or eGFP; see Fig. S1b for detailed description). As result, T-Sapphire positivity indicated cells expressing the shRNA targeting a control (shCTRL), while eGFP positivity indicated cells expressing the shRNA targeting the gene-of-interest (shGOI). If the GOI harbors an essential function, the eGFP-positive population gets lost over time in vivo, indicating that the patient might profit from drugs targeting the GOI. **b** Switch in fluorochrome expression upon Cre-ER[T2]-recombination: Double transgenic PDX AML-388 cells expressing Cre-ER[T2] together with either iRFP/shCTRL or mTagBFP/shGOI (sh*MCL1*) were mixed 1:1 and injected into the tail vein of NSG mice ($3 \times 10^5$ cells/mouse; $n = 14$). 7 days after injection, 2 mice were sacrificed and PDX cells analyzed by flow cytometry for all 4 fluorochromes. In the remaining mice, 50 mg/kg TAM was administered by oral gavage to induce Cre-ER[T2]-mediated recombination. Resulting increase in T-Sapphire or eGFP expression, indicating expression of shCTRL and shGOI, respectively, was measured in PDX cells isolated from mice at the indicated time points (24, 36, 52 and 72 h after TAM; $n = 3$ per time point). Representative histograms and plots are shown. **c** Typical results for a GOI with essential function: Upper scheme depicts the experimental procedure: For pairwise competitive assays, mice were injected with either of two mixtures: a control mixture of iRFP/shCTRL and mTagBFP/shCTRL (short shCTRL/shCTRL) or the experimental mixture iRFP/shCTRL and mTagBFP/shGOI (short shCTRL/shGOI); as GOI, the apoptosis regulator *MCL1* was chosen (shGOI = sh*MCL1*) ($3 \times 10^5$ cells/mouse, data from 4 exemplary mice are shown). TAM was administered 7 days after injection (day 0). Mice were sacrificed 3 and 26 days after TAM and PDX cells analyzed for expression of the inducible fluorochromes T-Sapphire and eGFP. Density plots show representative results for both mixtures on day 3 (left) and day 26 (right). Right panels show quantification as percentage of [eGFP/shGOI positive cells divided by (the sum of T-Sapphire/shCTRL and eGFP/shGOI positive cells)]; the shCTRL/shCTRL mixture is analyzed and depicted, respectively.

ALL samples showed minor to no effects on growth, proving patient-individual sensitivities (Figs. 2c and S2b). Silencing *MCL1* in AML-388 induced rapid cell death, which was already detectable within the first 72 h after TAM administration (Fig. S2c–e). Gene set enrichment analysis from RNA sequencing data comparing shCTRL and sh*MCL1* PDX cells indicated that *MCL1* knockdown was associated with activation of the apoptosis pathway, verified using Annexin-V staining (Fig. S2d–e). To visualize selective loss of individual GFP-positive cells upon *MCL1* silencing, re-transplantation experiments into wildtype zebrafish (danio rerio) were performed, which confirmed significant and rapid depletion of PDX cells upon *MCL1* knockdown between 48 and 72 h after TAM in an independent in vivo model (Fig. S2f).

Taken together, using the inducible knockdown approach, *MCL1* could be identified as a therapeutic vulnerability in one of 3 PDX samples, for which functional relevance could not be predicted by expression levels of anti-apoptotic BCL-2 family members, highlighting the need for functional assays (Fig. S2g).

As silencing of *MCL1* induced cell death in PDX AML-388, but not in ALL-199 nor ALL-265, we next asked whether this correlates with response towards pharmacological inhibition of *MCL1*. We studied the small molecule antagonist S63845 (Fig. 2d), which has previously been shown to be effective in AML cell lines and PDX samples[31,35,36] and is currently under clinical investigation as single agent (NCT02979366) or in combination regimens (NCT03672695). Treatment of mice bearing AML-388 significantly diminished tumor burden as monitored by in vivo bioluminescence imaging (Fig. 2e), reduced splenomegaly (Figs. 2f and S2h) and number of PDX cells (Fig. S2i) re-isolated from the murine spleens or bone marrow. In contrast, the *MCL1* inhibitor had no effect on ALL-199, recapitulating effects observed in the inducible knockdown system. Thus, the inducible knockdown system correlated to response of PDX samples to the pharmacological inhibition, confirming the use of this technique as surrogate to study sample-specific vulnerabilities on a molecular level in a highly clinically relevant setting.

Because *MCL1* has been shown to confer resistance to several anticancer drugs[37], we examined in a next step whether knockdown of *MCL1* strengthens the response of AML PDX

models towards drug treatment in vivo. Groups of mice were treated either with the BCL-2 inhibitor ABT-199 (Venetoclax) (Fig. 2g), or the conventional chemotherapeutic drug Cytarabine (Fig. S2j–k) at doses that do not significantly reduce tumor burden in mice. Both treatments further decreased the *MCL1* knockdown population in a synergistic way, indicating that sensitivity towards ABT-199 or Cytarabine might be increased by *MCL1* directed treatment in patients (Fig. 2g, Fig. S2j–k). Thus, using *MCL1* as exemplary target, we provide evidence that our approach enables distinguishing between subgroups of tumors in order to select patients, which might profit from therapies targeting a certain GOI, and to evaluate treatment combinations.

**Specific targeting of the fusion oncogene *MLL-AF4*.** To further validate the specificity of our approach, we next studied a bona fide positive control with high likelihood of harboring an essential function in established PDX tumors in vivo. The translocation t(4;11) and corresponding expression of the *MLL-AF4* fusion (KMT2A-AFF1) is present in 80% of infant B-precursor ALL patients, and is associated with poor prognosis[38]. Several studies elucidated its role in ALL cell lines and mouse models[39], but up to date no molecular investigations on its function have been carried out in patient cells or established tumors growing in vivo. We designed a shRNA targeting a mRNA breakpoint shared by several patients, which significantly reduced expression of the fusion transcript (Fig. 3a). Because the shRNA sequence targeted neither of the individual wildtype genes, MLL or AF4 (Figs. 3a and S3a–b), no major adverse effects on normal tissue are expected when applied in vivo, e.g., by systemic gene therapeutic approaches. Inducible knockdown of *MLL-AF4* significantly reduced ALL cells in the t(4;11)-positive PDX model tested, but not in a translocation-negative sample, proving a tumor maintaining role of *MLL-AF4* in established patient tumors in vivo (Figs. 3b and S3a). Variations between the different animals were neglectable reflecting the high reliability of our approach (Fig. 3b). Reduced tumor growth of the sh*MLL-AF4* mixture was visible using in vivo imaging, even though 50% of injected tumor cells expressed shCTRL (Fig. 3c). Gene expression analysis demonstrated that shCTRL cells expressed a set

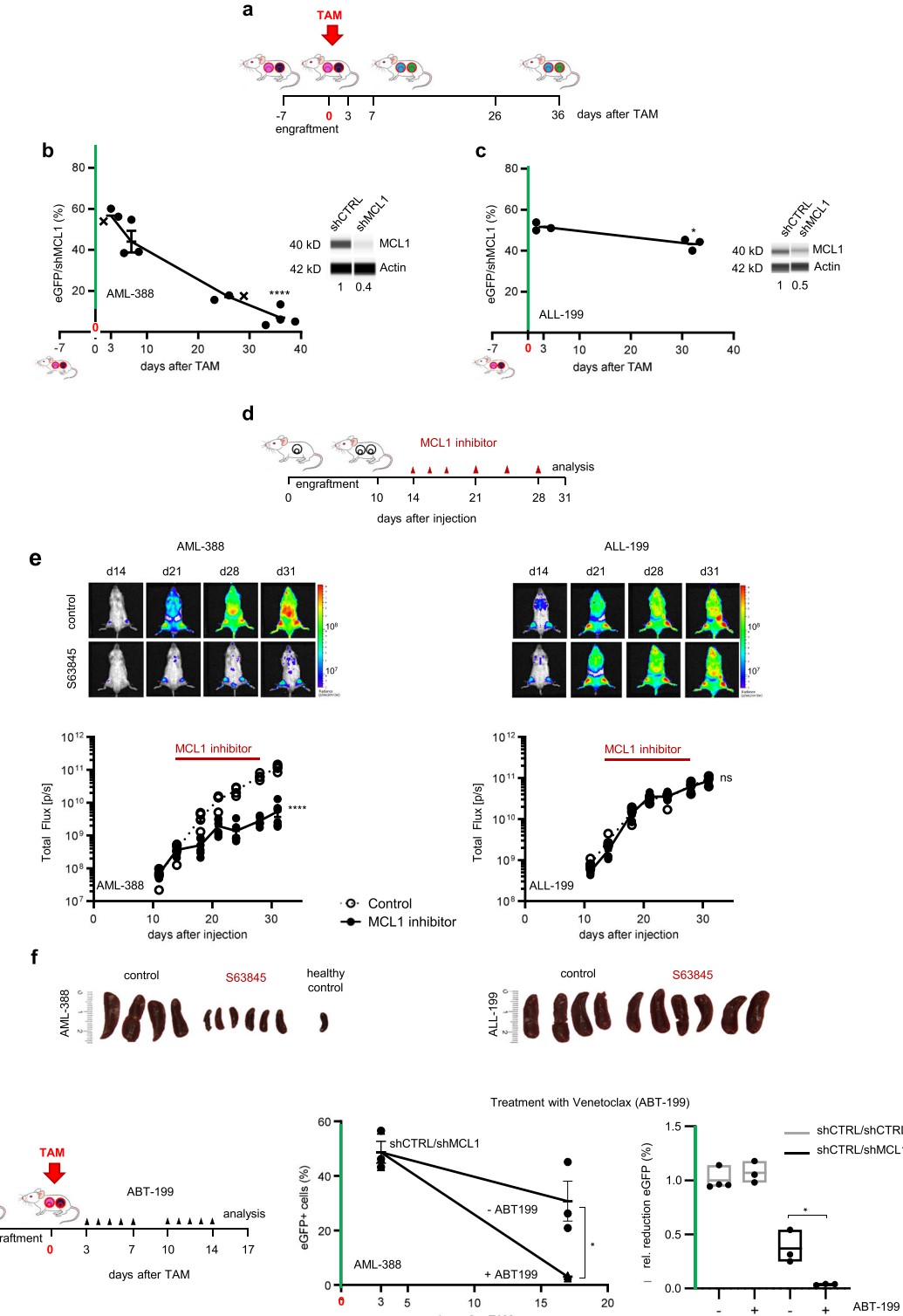

of genes characteristic for samples with the *MLL-AF4* translocation[40], which was no longer present upon sh*MLL-AF4* knockdown, where an expression signature similar to non-MLL rearranged samples prevailed (Fig. S3c–d).

These results prove the selectivity and operability of our technique and showed that *MLL-AF4* harbors an essential function in established patient-derived leukemias growing in vivo. We provide strong molecular evidence in a clinically relevant model that the translocation transcript represents an attractive therapeutic target for future therapies.

**DDIT4L is a therapeutic vulnerability in *DUX4-IGH* rearranged acute lymphoplastic leukemia.** In a last step, we examined a less well studied tumor alteration, the recently discovered rearrangement t(4;14) which occurs in 7% of ALL patients and results in the *DUX4-IGH* gene fusion[41]. Because cells with t(4;14) display high levels of otherwise absent *DUX4*, we asked whether *DUX4* represents a vulnerability in this subgroup of ALL in vivo. Using our technique, we demonstrated an essential function for *DUX4* in t(4;14) rearranged ALL-811 (Fig. 3d). Expression of the *DUX4-IGH* translocation was reported to be associated with a defined gene expression

**Fig. 2 Inducible knockdown of *MCL1* in vivo predicts response of AL PDX to pharmacological *MCL1* inhibition. a–c** Inducible knockdown of *MCL1* in AL PDX. **a** Scheme depicting the experimental setup. Groups of mice were injected with the shCTRL/shMCL1 mixture for competitive in vivo assays (3×10⁵ cells/mouse). TAM was administered when tumors were established; differences between eGFP-positive sh*MCL1* cells among all recombined cells were determined 3 days after TAM and at end stage leukemia to assess essentiality of *MCL1*. **b–c** Competitive experiments were set up as described in **a**; TAM (50 mg/kg) was applied once (day 0); mice bearing (**b**) AML-388 PDX cells were sacrificed 3 ($n = 3$), 7 ($n = 3$), 26 ($n = 3$) and 36 ($n = 4$) days after TAM; mice bearing ALL-199 PDX cells were sacrificed 3 ($n = 3$) and 32 ($n = 3$) days after TAM. *MCL1* protein expression was analyzed in sorted shCTRL and sh*MCL1* populations by protein immunoassay (Simple Western) 7 days after TAM (AML-388) or at the experimental endpoint (ALL-199). Mean ± SEM of the proportion of eGFP-positive cells isolated out of all recombined cells is displayed; each dot represents one mouse; x marks mice shown in Fig. 1c. To determine significance of depletion of shGOI-expressing cells, the percentage of eGFP/shGOI cells at the experimental endpoint is compared to the percentage of eGFP/shGOI cells at 3d post TAM, as this time point is used to define the sample-specific recombination efficiency. *$p = 0.0136$, ****$p < 0.0001$, ns not significant by unpaired *t*-test. **d–f** Pharmacological inhibition of *MCL1* in AL PDX. **d** Scheme depicting the experimental setup. Groups of mice were injected with AML-388 (left; 3×10⁵ cells/mouse, $n = 10$) or ALL-199 (right; 1×10⁶ cells/mouse, $n = 10$) PDX cells expressing firefly luciferase. 14 days after injection, mice were treated with the small molecule *MCL1* antagonist S63845 (mice received 25 mg/kg three times in the first week, 12.5 mg/kg twice in the second week, and once in the third week, $n = 6$) or solvent as control ($n = 4$) and tumor growth was monitored by bioluminescence in vivo imaging until mice were sacrificed 31 days after injection. **e** Representative bioluminescence images are depicted and graph shows mean ± SEM; ****$p < 0.0001$, ns not significant by unpaired *t*-test. **f** Images of spleens of control- or S63845-treated mice are displayed. One spleen of a healthy mouse without leukemia (healthy control) is shown for comparison. **g** The combinatorial effect of *MCL1* knockdown plus ABT-199 (Venetoclax) was studied by injecting mice with a 1:1 mixture of shCTRL/sh*MCL1* AML-388 cells (3×10⁵ cells/mouse) and treating them with 50 mg/kg TAM, 7 days after injection (day 0). 3d after TAM administration, control mice were sacrificed ($n = 3$) and the remaining mice treated either with 100 mg/kg ABT-199 ($n = 3$) or solvent ($n = 3$) for 5 consecutive days per week, in 2 cycles. At the end of the experiment (17 days after TAM), mice were sacrificed and analyzed as in Fig. 1c. Mean ± SEM is shown; *$p = 0.0194$ by unpaired *t*-test. Reduction of eGFP-positive cells in the shCTRL/sh*MCL1* mix relative to shCTRL/shCTRL (+/- ABT-199) is displayed. Each dot represents one mouse. Mean ± SEM is shown; *$p = 0.0194$ by unpaired *t*-test.

signature, previously referred to as the "ERG subtype"[42–46]. We performed gene expression analysis of sh*DUX4* and shCTRL ALL-811 cells (Fig. 3e) and performed gene set enrichment analysis (GSEA) with two published datasets[43,45]. We found genes over-expressed in *DUX4* knockdown NALM6 cells[43] also enriched in our sh*DUX4* PDX sample (Figs. 3f and S3e Set 1). Accordingly, genes downregulated in *DUX4* knockdown NALM-6 cells[43] (Fig. S3e, Set 2) and genes highly expressed in the cluster of patients characterized by *DUX4* translocation and ERG deletion[45] (Fig. S3f, Set 3) were enriched in the shCTRL sample (Fig. S3g). These data confirm the presence of the typical *DUX4* signature in shCTRL PDX cells and demonstrate reversal of this signature upon *DUX4* knockdown in a PDX model in vivo (Figs. 3e–f and S3e–g). Our technique could thus identify *DUX4* as attractive therapeutic target to treat the recently detected subgroup of *DUX4-IGH* rearranged ALL.

To further confirm the relevance of the detected genes for tumor maintenance of *DUX4*-rearranged samples we tested the role of one gene that was downregulated upon *DUX4* silencing in PDX ALL-811 and in NALM-6 cells (Fig. 3g), the DNA-damage-inducible transcript 4-like (*DDIT4L*; also known as Redd2 or Rtp801L), which has been shown to regulate mTOR signaling and autophagy in mammalian cells. *DDIT4L* expression is induced in the presence of different types of pathological stress, suggesting a possible involvement of *DDIT4L* in stress response[47–49]. Interestingly, we found *DDIT4L* highly expressed in *DUX4* rearranged ALL (Fig. S3h). Inducible knockdown of *DDIT4L* significantly diminished leukemic growth within 2 weeks of in vivo tumor growth (Fig. 3h–i), suggesting that downregulation of *DDIT4L* might have mediated, at least in part, the growth inhibitory effects observed in the sh*DUX4* population. Taken together, we identify *DDIT4L* as a therapeutic vulnerability in the *DUX4-IGH* subtype of B-ALL.

## Discussion
We have established a method which combines an in vivo approach with patient-derived tumor cells and pre-established tumors for inducible knockdown and allows validating vulnerabilities on an individual patient level. We established the technique, as preclinical molecular approaches are lacking which faithfully mimic the situation of treatment in patients, characterized by existence of an established tumor in vivo. Our method is capable to evaluate the functional relevance of tumor alterations (i) in the background of

individual patient tumors and their specific characteristics; (ii) in the complex in vivo environment of living beings and; (iii) in the situation of a pre-existing tumor, avoiding influences irrelevant for patients. Our molecular approach closely mimics the clinical situation and complements an important step in the evaluation chain of precision oncology. The molecular technique allows target validation, for single agent use or in combination therapies, independently from confounders such as pharmacodynamics and pharmacokinetics, toxicity and lack of specificity, inherent to drugs and compounds[50]. Inducible genetically engineered mouse models (GEMM) allow studying gene function independently from, e.g., gestation-specific processes; in analogy, our approach allows studying vulnerabilities devoid of model-inherent processes like in vitro culture, transplantation, homing and engraftment. Our inducible approach closely controls for putative clonal bias as identical cells are studied, before and after induction of knockdown. Our knockdown approach might complement CRISPR/Cas9-mediated knockout approaches[16], while putatively more coherently mimicking the partial, but incomplete target inhibition induced by drugs or compounds. In addition to alterations detected by sequencing efforts, our technique allows functional evaluation of targets detected by sequencing-agnostic approaches, e.g., in cell death pathways, and studying un-druggable targets, including non-coding RNAs[51].

While we studied acute leukemias as model diseases, the CRE-loxP-system has been successfully used in numerous different tumor entities and our technique can easily be transferred to other cancers. We envision a major potential of our method on a proof-of-concept level, where deeper knowledge on tumor dependencies will improve drug design and the ability to interpret patient sequencing data. It might also serve as a highly clinic-related, functional biomarker to improve clinical decision making to individualize treatment. Due to its major potential to tailor drug development, improve patient care and increase the success rate of clinical trials, our technique will foster personalized oncology in the future.

## Methods

**Ethical statement.** Written informed consent was obtained from all patients and from parents/carers in the cases where patients were minors. The study was performed in accordance with the ethical standards of the responsible committee on human experimentation (written approval by Ethikkommission des Klinikums der Ludwig-Maximilians-Universität München, Ethikkommission@med.unimuenchen.de, April 15/2008, number 068-08; September 24/2010, number 222-10; January 18/2019, number

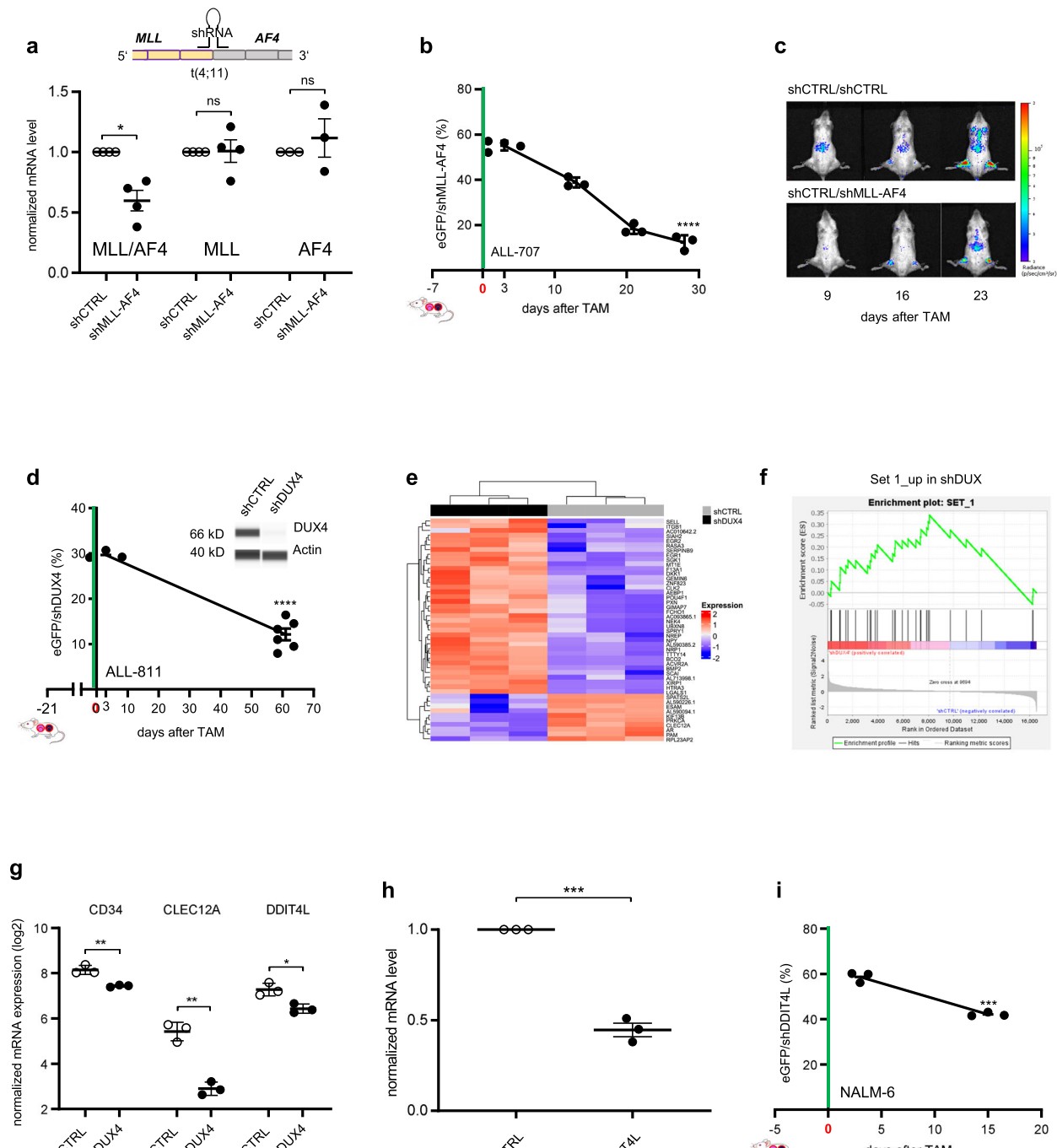

222-10) and with the Helsinki Declaration of 1975, as revised in 2000. All animal trials were performed in compliance with the ARRIVE guidelines (https://arriveguidelines.org) and in accordance with the current ethical standards of the official committee on animal experimentation (written approval by Regierung von Oberbayern, tierversuche@reg-ob.bayern.de, January 15/2016, Az. ROB-55.2Vet-2532.Vet_02-16-7; Az. ROB-55.2Vet-2532.Vet_02-15-193; ROB-55.2Vet-2532.Vet_03-16-56).

**Animal model.** Six to 16 weeks old male and female NOD.Cg-Prkdc[scid] IL2rg[tm1Wjl]/SzJ (NSG) mice (The Jackson Laboratory, Bar Harbour, ME, USA) were included. Mice were kept under specified pathogen-free (SPF) conditions with a 12/12 h light cycle, temperature of 20–24 °C and 45–65% humidity according to Annex A of the European Convention 2007/526 EC. The maximum stocking density of the cages corresponds to Annex III of the 2010/63 EU. The cages were constantly filled with structural enrichment and the animals had unlimited access to food and water. During the experiment, mice were kept in individually ventilated cages (IVCs). Hygiene monitoring was carried out at least quarterly in accordance with the current FELASA recommendation.

Donor mice used for PDX cell amplification were sacrificed at advanced leukemic disease (more than 60% leukemic cells within peripheral blood) or when first clinical signs of illness appeared (rough fur, hunchback, reduced motility, paralysis). Experimental mice were sacrificed at specified time points.

**Generating transgenic patient derived xenograft (PDX) models.** Establishing serially passaged AML and ALL PDX models in NSG mice, re-isolating PDX cells from mice, PDX cell culture, lentiviral transduction, enrichment of transgenic cells and in vivo imaging were described previously[29,52,53].

**Generation of AML and ALL-PDX models.** Fresh primary AML or ALL cells were isolated by Ficoll gradient centrifugation from peripheral blood or bone marrow aspirates that had been obtained from leftovers of clinical routine sampling before onset of therapy and injected into 6–12 weeks old NSG mice via the tail vein. Engraftment was monitored by 2-weekly flow cytometry measurement of human cells in peripheral blood starting at week 4. Mice were sacrificed at first clinical signs of disease, as measured by quantification of human cells in peripheral blood. From engrafted mice

**Fig. 3 Essential function of *MLL-AF4* and *DUX4-IGH* fusion proteins in rearranged ALL. a–c** *MLL-AF4* plays an essential role in vivo in *MLL-AF4* rearranged ALL. **a** A shRNA targeting the *MLL-AF4* fusion mRNA was designed, according to the patient's specific breakpoint of PDX ALL-707 (Table S3). mRNA expression of *MLL-AF4*, MLL and AF4 in PDX ALL-707 was analyzed by qPCR in CTRL and *MLL-AF4* knockdown cells ($n = 3$ each). Mean ± SEM of cells isolated from mice 28 days after TAM are shown. *$p = 0.0178$ by Welch's *t*-test; ns not significant. **b** Competitive experiments were performed and analyzed as in Fig. 1c, using PDX ALL-707 cells and the shGOI targeting *MLL-AF4*. TAM was applied on two consecutive days (100 mg/kg, day $-1 + 0$). Mice were sacrificed 3 ($n = 4$), 13 ($n = 3$), 21 ($n = 3$) and 28 ($n = 3$) days after TAM; each dot represents one mouse; mean ± SEM; ****$p < 0.0001$, by unpaired *t*-test. **c** Representative in vivo bioluminescence images of mice bearing a shCTRL/shCTRL or shCTRL/sh*MLL-AF4* mixture from the experiment described in Fig. 3b, at the indicated time points after TAM administration. **d–g** *DUX4* plays an essential role in *DUX4-IGH* rearranged ALL. **d** Competitive experiments were performed and analyzed as in Fig. 2b, using ALL-811 and the shRNA targeting *DUX4* ($1.4 \times 10^6$ cells/mouse). 21 days after injection, TAM (50 mg/kg) was applied once (day 0). Mice were sacrificed 3 ($n = 3$) and 61 ($n = 6$) days after TAM. Shown is mean ± SEM. ****$p < 0.0001$ by unpaired *t*-test. Protein immunoassay of *DUX4* in NALM-6 cells, after lentiviral transduction with the indicated shRNAs. β-actin was used as loading control. **e** Transcriptome analysis was performed from eGFP/shCTRL and eGFP/sh*DUX4* cells from the experiment described in panel **d** 82 days after TAM ($n = 3$ for each condition). Heatmap of 47 genes differentially expressed between the two groups is shown. All gene expressions have been scaled to a mean value of 0 and a variance of 1. **f** Enrichment plot of genes deregulated in sh*DUX4* PDX cells compared to genes upregulated two-fold (Set 1) in a published transcriptomic signature (Tanaka et al.[43]) generated from NALM-6 cells expressing sh*DUX4*. NES = 2.19 (FDR *q*-value < 0.002). **g** Mean ± SEM ($n = 3$ independent animals for shCTRL or shDUX4) of three differentially expressed genes are depicted; **$p = 0.0040$ for CD34, **$p = 0.0011$ for CLEC12A and *$p = 0.0145$ for DDIT4L by unpaired *t*-test. **h, i** *DDIT4L* inhibition partially phenocopies *DUX4* silencing. **h** mRNA expression of *DDIT4L* in NALM-6 was analyzed by qPCR in CTRL and *DDIT4L* knockdown cells ($n = 3$ each). Mean ± SEM of cells isolated 7 days after TAM are shown. ***$p < 0.001$ by unpaired *t*-test. **i** Competitive experiments were performed and analyzed as in Fig. 2b, using the NALM-6 cell line and the shRNA targeting *DDIT4L* ($5 \times 10^6$ cells/mouse (for day 3) and $1 \times 10^5$ cells/mouse (for day 15)). 5 days after injection, TAM (50 mg/kg) was applied once (day 0). Mice were sacrificed 3 ($n = 3$) and 15 ($n = 3$) days after TAM. Shown is mean ± SEM. ***$p = 0.0003$ by unpaired t-test.

---

(first generation), PDX AML or ALL cells were reisolated out of femurs, tibiae and spleen by mincing the tissues and filtration through a cell strainer, followed by Ficoll gradient centrifugation in case of splenic cells[29]. PDX AML cells were identified by staining for human CD45, CD33, CD3 and CD19 (CD38 for PDX ALL) and flow cytometry analysis. Without further enrichment or manipulation, $1 \times 10^6 - 5 \times 10^6$ total BM cells were reinjected into next recipient NSG mice for reexpansion (secondary transplantation).

**Lentiviral transduction and cell enrichment.** Lentiviral transduction was performed as previously described[52]. Briefly, PDX cells freshly isolated from mouse spleen or BM were re-suspended in RPMI-Medium (Life Technologies) supplemented with 20% fetal calf serum (Biochrom AG, Berlin, Germany), 5% L-Glutamin, 1% Gentamycin, 1% Penicillin/Streptomycin, 0.6% mixture of rh insulin/ human transferrin/sodium selenite (Life Technologies), 1 mM sodium pyruvate, and 50 μM 1-thioglycerole (Sigma-Aldrich, Hannover, Germany). $1 \times 10^6$ cells in 1 ml medium were transferred to a cell culture plate and were transduced overnight with lentiviral constructs in the presence of 8 μg/ml polybrene (Sigma-Aldrich). To save one round of passaging through mice, PDX cells freshly transduced with lentiviruses were kept in culture for 4 days to allow marker expression and enrichment of transgenic cells using a FACSAria III (BD Bioscience) and the FACSDiva software 8.0.2 (BD Bioscience).Sorted cells were then re-injected into next generation recipient mice.

**Bioluminescence in vivo imaging.** In vivo bioluminescence imaging (BLI) BLI was performed as previously described[52]. The IVIS Lumina II Imaging System was used (Caliper Life Sciences, Mainz, Germany). Mice were anesthetized using isoflurane, placed into the imaging chamber in a supine position and fixed at the lower limbs and by the inhalation tube. Coelenterazine (Synchem OHG, Felsberg/Altenburg, Germany) was dissolved in acidified methanol (HPLC grade) at concentration 10 mg/ml and diluted shortly before injection in sterile HBG buffer (HEPES-buffered Glucose containing 20 mM HEPES at pH 7.1, 5% glucose w/v). Immediately after intravenous tail vein injection of 100 μg of native Coelenterazine, mice were imaged for 15 s using a field of view of 12.5 cm with binning 8, f/stop 1 and open filter setting. To monitor tumor growth, mice were imaged once weekly; after therapy, mice were imaged every other day.

**Quantification of BLI pictures.** Quantification of BLI signal was performed as previously described[52]. The Living Image software 4.4 (Caliper Life Sciences, Mainz, Germany) was used for data acquisition and quantification of light emission using a scale with a minimum of $1.8 \times 10^4$ photons per second per cm2 per solid angle of one steradian (sr). Different regions of interest (ROI) were defined and signals were considered positive, when light emission exceeded background in each ROI. Background was measured in mice harboring GLuc negative leukemias. A ROI covering the entire animal was used (background $4 \times 10^6$ photons per second). As an exception to determine early engraftment, a small ROI covering the femurs was used (background $6 \times 10^4$ photons per second), as light emission became visible there first. Overt leukemia was considered above $10^{10}$ photons per second using the ROI covering the entire animal.

**Cloning.** For constitutive expression of the Cre-ER^T2 recombinase, the coding sequence of the enzyme was PCR amplified from the CreERT2FrtNeoFrt cassette (gift from MSS)

using a 5' primer carrying NsiI and a 3' primer carrying P2A-NsiI and ligated into the NsiI digested pCDH-SFFV-GLuc-T2A-mCherry vector downstream of the T2A peptide (Fig. S2a) (pCDH-vector, System Bioscience). For inducible knockdown of target genes, the lentiviral FLIP vector system[25,26] was optimized to link shRNA expression to fluorochrome expression. We used the lentiviral pCDH backbone, digested the vector with SpeI and SalI and introduced the following elements as a pre-synthetized stretch of DNA (GenScript®, Piscataway, NJ, USA): *SpeI* - SFFV - lox2272 - mTagBFP (iRFP720) - lox5171 - mir30 cassette-eGFP (T-Sapphire) -lox2272 - lox5171 – *SalI* (Fig. S2b). The shRNA sequences targeting the different genes (*MCL1*, *DUX4*, *DDIT4L*; see Table S2) were designed using the SplashRNA algorithm[54], with the exception of *MLL-AF4* where sequences were designed to directly cover the patient-specific translocation breakpoint (Table S2). As control, a shRNA targeting the Renilla luciferase was used in all experiments (shCTRL). The shRNA-sequences were introduced into the miR30 cassette of the KD vector as part of pre-synthetized and annealed, complementary single strand DNA oligos (110 bps, see Table S2; Integrated DNA Technologies, USA), having XhoI and EcoRI as 5' and 3' restriction sites, respectively. For knockdown of *MLL-AF4*, the miR-E KD cassette was used[55] and concatemerized to enhance the knockdown efficiency[56].

**In vivo assays and Tamoxifen administration.** For pairwise competitive in vivo experiments, PDX transduced with either the control shRNA expressing iRFP (iRFP720) or the shRNA against the GOI expressing mTagBFP, were freshly isolated from a donor mouse, were mixed at a 1:1 ratio (shCTRL/shGOI mix) and cells were injected into the tail vein of recipient NSG mice. Of note, to achieve reliable and reproducible results, the use of PDX cells freshly isolated from donor mice (not frozen/ thawn cells) is recommended. At best, the initial mixture should not substantially differ from a 1:1 mix. As a control, several groups of mice were injected with the shCTRL/ shCTRL mix, consisting of PDX cells transduced with either the control shRNA expressing iRFP or the control shRNA expressing mTagBFP. To promote Cre-ER^T2 translocation to the nucleus and induction of RNA interference, Tamoxifen (TAM, Cat#T5648-5G, Sigma) was resuspended in a sterile mixture of 90% corn oil (Cat#C8267-500ML, Sigma) and 10% ethanol at final concentration of 20 mg/ml; aliquots were stored for a maximum of 3 months at −20 °C. Before administration to mice, the solution was heated to 37 °C and applied via oral gavage. TAM concentrations were titrated to induce substantial shRNA expression and was given once at 50 mg/kg for AML-388, ALL-199, ALL-265 and ALL-811, while animals with ALL-707 received 100 mg/kg TAM on two consecutive days. TAM was given by earliest 7 days after cell transplantation and after engraftment was completed.

**Flow cytometric analysis of competitive in vivo experiments.** Freshly isolated PDX cells were analyzed using LSRII (BD Bioscience) to determine fluorochrome distributions. Forward/Side scatter analysis was used to gate on living cells, followed by gating on mCherry (Cre-ER^T2) positive PDX cells. At the beginning, the two cell populations of the mixture were distinguished by expression of either iRFP or mTagBFP. Upon Cre-ER^T2 recombination, cells expressing shCTRL started expressing T-Sapphire (instead of iRFP), while cells expressing shGOI expressed eGFP (instead of mTagBFP) (Fig. S1b); the color switch was monitored in two separate histograms for either T-Sapphire or eGFP (Fig. 1b). The final analysis combined and compared all cells expressing either of the two shRNAs, either T-Sapphire/shCTRL or eGFP/shGOI (Fig. 1b and c).

To determine the sensitivity of different PDX samples to inhibition of selected GOI, the percentage of cells with knockdown of the GOI (eGFP-expressing cells) were compared between starting conditions (3 days after TAM) to later time points, using at least $n = 3$ data points per time point and condition. A significant depletion in the amount of eGFP/shGOI positive cells over time characterized PDX samples sensitive to the knockdown of the GOI. For target genes inducing rapid cell death upon knockdown, day 1 after TAM administration can be used for comparison. To separate shCTRL and shGOI populations for further investigations, cells were sorted using FACSAria III (BD Bioscience).

**Statistical analysis**. Statistical significance of pairwise competitive in vivo experiments was analyzed by comparing the percentage of eGFP-positive cells out of all recombined cells (sum of T-Sapphire positive plus eGFP positive cells) between the shCTRL/shGOI mix at 72 h after TAM administration with the shCTRL/shGOI mix at the end of each experiment. Statistical analyses were performed using GraphPad Prism 8. Student's *t*-test was used, if not differently stated in the legends. A p-value of ≤0.05 was considered significant.

**In vivo drug treatment**. For in vivo treatment with ABT-199 (Venetoclax, SelleckChem, USA) or Cytarabine (Cell Pharma GmbH, Bad Vilbel, Germany), mice were injected with a 1:1 mixture of shCTRL/sh*MCL1* AML-388 PDX cells ($3 \times 10^5$ cells/mouse) and TAM was administered one week thereafter to all animals. 72 h after TAM, three mice were sacrificed to determine recombination efficiency. The remaining animals were divided into three groups and treated either with solvent ($n = 3$) or ABT-199 (100 mg/kg in Carboxymethyl cellulose (1% w/v) + DMSO (2% v/v) by oral gavage for 5 consecutive days and 2 weeks; $n = 3$) or Cytarabine (100 mg/kg in PBS by intraperitoneal injection for 4 consecutive days and 1 week; $n = 3$). At the end of the experiment, mice were sacrificed, BM processed and PDX cells analyzed by flow cytometry for subpopulations' distribution.

Synergistic effect was calculated using the fractional product method[57]. Measured survival rates were 0.39 upon *MCL1* KD and 1.0 upon Venetoclax; expected apoptosis induction of independent application of *MCL1* knockdown and Venetoclax was calculated as [(1 minus (survival after simulation with *MCL1* knockdown) times (survival after stimulation with VCR)) times 100] which resulted to be 0.61; measured apoptosis by the combination of *MCL1* and Venetoclax was 0.94 and thus much higher than the expected apoptosis of 0.61, proving that the combination acted in a synergistic way.

For in vivo treatment with S63845 (Hölzel Diagnostika, HY-100741-50mg), mice were injected with luciferase expressing ALL-199 ($1 \times 10^6$ cells/mouse) or AML-388 PDX cells ($3 \times 10^5$ cells/mouse). Tumor growth was monitored twice per week by bioluminescence imaging. Two weeks after cell injection, mice were treated with S63845 (12.5 mg/kg in 25 mM HCl + 20% 2-hydroxy propyl β-cyclo dextrin by i.v. injection; week 1: 3 doses; weeks 2 and 3: two doses). At sign of overt leukemia, mice were sacrificed, spleens weighted and the proportion of PDX cells in BM and spleen analyzed by flow cytometry.

**Engraftment of PDX cells in zebrafish**. For PDX cell preparation, AML-388 PDX cells expressing (i) mCherry-Cre-ER$^{T2}$ and (ii) the knockdown construct mTagBFP/sh*MCL1*, were amplified in a donor mouse. Mice were sacrificed, human cells isolated and treated in vitro with 50 nM TAM (Sigma-Aldrich, H7904-25G) to induce recombination and shRNA expression. To allow competitive experiments comparing cells with and without recombination, mCherry positive cells were sorted 48 h after TAM to gain a 1:1 mixture of eGFP:mTagBFP positive cells and thus 50% of cells with Cre-ER$^{T2}$-induced recombination.

48 h after fertilization, dechorionated, 1-phenyl 2-thiourea (PTU) treated (75 µM) (Sigma-Aldrich, P7629) wild type zebrafish embryos (Danio rerio, AB line) were anesthetized with Tricain (Sigma-Aldrich, A5040). Embryos were injected through the Duct of Cuvier, using a Femtojet microinjector (Eppendorf, Hamburg, Germany), with 200 to 500 AML-388 PDX cells per embryo of the mTagBFP/sh*MCL1* mixture. Embryos were raised at 36 °C. At 4 and 28 h post transplantation (hpt), embryos were anesthetized with 750 µM Tricain and embedded in 1.5% low melting-temperature Agarose (Lonza, MetaPhor Agarose 50185) containing 75 µM PTU and 750 µM Tricaine.

Each larva was imaged using a spinning disc microscope (20x magnification) and images were applied to maximal intensity projection. Using the spot detection function (LoG detector) of the Image-J plugin TrackMate[58] PDX cells were identified by mCherry-Cre-ER$^{T2}$ expression. To quantify the subfraction of cells expressing the shRNA, the median eGFP signal was determined at 4 hpt. For each fish the percentage of eGFP positive, shRNA expressing cells was calculated at 4 hpt and 28hpt using the determined median as threshold.

Zebrafish embryos/larvae were studied exclusively within the first 5 days after fertilization, handled compliant to local animal welfare regulations and maintained according to standard protocols (www.ZFIN.org) which does not require a special permit according to German Laboratory Animal Protection Law.

**Flow cytometric analysis of BH3 proteins' level and Annexin V staining**. To determine intracellular expression levels of BH3 proteins, cells were fixated in 2% paraformaldehyde, permeabilized using perm/wash buffer (BD Bioscience, Franklin Lakes, NJ, USA) and subsequently stained with fluorescently labeled antibodies against BCL-2 (clone Bcl-2/100, BD Bioscience), BCL-XL (clone 54H11, Cell Signaling, Cambridge, UK), MCL-1 (Clone D2W9E, Cell signaling) or respective isotype controls (Cat.: 556357, BD Bioscience; clone DA1E, Cell Signaling). Dead cells were excluded by Fixable Viability Dye staining. If not otherwise stated, reagents and antibodies were purchased from eBioscience. Flow cytometric analysis was performed on a BD FACS Canto II (BD Bioscience) and data were analyzed using FlowJo software (TreeStar Inc., Ashland, OR, USA).

Annexin V staining was performed on PDX AML-388, ALL-199 and ALL-265 cells isolated from the mouse BM 72 h after TAM treatment or thawed and treated in vitro, using PE/Cy7 Annexin V (BioLegend, 640949) according to the manufacturer's instruction and analyzed by flow cytometry (LSRII, BD Bioscience).

**Targeted genome sequencing**. The *MLL-AF4* breakpoint was sequenced at the certified laboratory for Leukemia Diagnostics, Department of Medicine III, University Hospital, LMU Munich, Munich, Germany.

**Real-time quantitative PCR**. Total RNA from flow cytometry enriched populations was extracted using RNeasy Mini Kit (Qiagen, Venlo, Netherlands) and reverse transcribed using the QuantiTect Reverse Transcription kit (Qiagen, Venlo, Netherlands) according to manufacturer's instruction. Quantitative PCR was performed in a LightCycler 480 (Roche, Mannheim, Germany) using the corresponding LightCycler 480 Probes Master and the pre-designed Probes of the Universal ProbeLibrary (Roche, Mannheim, Germany). The primer and probes used for qPCR are: *HPRT1*_fw: TGATAGATCCATTCCTATGACTGTAGA, *HPRT1*_rv: CAAGACATTCTTTCCAGTTAAAGTTG, UPL #22; *MLL/AF4*_fw: AAGTTCCCAAAACCACTCCTAGT, MLL/AF4 rv: GCCATGAATGGGTCAT TTCC, UPL #22; *MLL*_fw: AAGTTCCCAAAACCACTCCTAGT, *MLL*_rv: GATCCTGTGGACTCCATCTGC, UPL #22; *AF4*_fw: CTCCCCTCAAAAAG TGTTGC, *AF4*_rv: TAGGTCTGCTCAACTGACTGAG, UPL #84; *DDIT4L*_fw: CCCAGAGAGCCTGCTAAGTG, *DDIT4L*_rev: TTGCTTTGATTTGGACAGA CA, UPL #67. Relative gene expression levels were normalized to *HPRT1* using the $2^{-\Delta\Delta Ct}$ method.

**Gene expression profiling**. Gene expression analysis was performed by applying a bulk adjusted SCRB-Seq protocol on sorted subpopulations from PDX samples as described previously[59,60]. Briefly, for library preparation 2,000 cells of each individual sample were sorted and lysed in RLT Plus (Qiagen) supplemented with 1% 2-Mercaptoethanol (Sigma–Aldrich) and stored at −80 °C until processing. A modified SCRB-seq protocol (6, 7) was used for library preparation. Briefly, proteins in the lysate were digested by Proteinase K (Ambion), RNA was cleaned up using SPRI beads (GE, 22% PEG). In order to remove isolated DNA, samples were treated with DNase I for 15 min at RT. cDNA was generated by oligo-dT primers containing well specific (sample specific) barcodes and unique molecular identifiers (UMIs). Unincorporated barcode primers were digested using Exonuclease I (Thermo Fisher). cDNA was pre-amplified using KAPA HiFi HotStart polymerase (Roche) and pooled before Nextera libraries were constructed from 0.8 ng of pre-amplified cleaned up cDNA using Nextera XT Kit (Illumina). 3' ends were enriched with a custom P5 primer (P5NEXTPT5, IDT) and libraries were size selected using 2% E- 6 Gel Agarose EX Gels (Life Technologies), cut out in the range of 300–800 bp, and extracted using the Monarch DNA Gel Extraction Kit (New England Biolabs) according to manufacturer's recommendations.

All raw fastq data was processed with zUMIs[61] (2.4.5b). Mapping was performed using STAR 2.6.0a[62] against the concatenated human (hg38) and mouse genome (mm10). Gene annotations were obtained from Ensembl (GRCh38.84/GRCm38.75). Analysis of RNA sequencing data followed standard recommendations[63]. Statistical analysis was performed using the R 3.6.1 software package (R Core Team, 2019). In case of multiple testing, p-values were adjusted using the Benjamini-Hochberg procedure (FDR-cutoff <0.05). Gene Set Enrichment Analysis (GSEA) using default settings (version 4.0.2) was used for the association of defined gene sets with different subgroups[64].

For PDX-707 Massice Analysis of cDNA Ends (MACE) was performed at GenXPro (Frankfurt am Main, Germany). Therefore, 28 days after TAM 50,000 cells eGFP/shCTRL ($n = 3$) and eGFP/sh*MLL-AF4* PDX ($n = 3$) cells were sorted and sent to GenXPro for total RNA isolation, MACE library preparation and strand-specific sequencing using the HiSeq2500 (Illumina, USA), as previously described[65]. The bioinformatic analysis was conducted in accordance to the analysis pipeline for MACE libraries by GenXPro GmbH. Distinct Oligo IDs and UMIs on each transcript enabled initial demultiplexing and subsequent removal of PCR-duplicates for alignment of adapter-free sequences with Bowtie 2 to the human reference genome (Genome Reference Consortium Human Build 38 patch release 13, GRCh38.p13). Considering sequencing depth and RNA composition, the sequencing data was normalized with the median of ratios method by DESeq2. GSEA was carried out to compare the effect of the *MLL-AF4* KD in the t(4;11) PDX ALL-707 with published transcriptomic data from t(4;11) leukemia patients (expression data from Stam et al.[40]; GEO database: GSE19475; significant genes were selected according to Lin et al. (2016): p ≤ 0.05, FDR ≤ 0.1, fold change ≥2). The GSEA software of UC San Diego and the Broad Institute was used for analysis. Permutation testing was conducted with a gene set specific permutation test, set to 1000 permutations.

To study *DUX-4* expression in B-ALL patients we downloaded log2-FPKM values of 1988 patients with B-progenitor ALL from the publicly available St. Jude

Cloud (https://viz.stjude.cloud/stjude/visualization/pax5-driven-subtypes-of-b-progenitor-acute-lymphoblastic-leukemia-t-sne~15), as previously described[66].

**Protein immunoassay**. To quantify protein of low PDX cell numbers, the Simple Western capillary protein immunoassay (WES, ProteinSimple, San Jose, USA) was performed according to manufacturer's instructions as previously described[67]. Flow cytometry enriched cell populations were incubated in lysis buffer (#9803, Cell Signaling Technology, Boston, USA) on ice for 30 min and protein concentration measured by BCA assay (#7780, New England Biolabs, Beverly, USA). Results were analyzed using the Compass software (ProteinSimple). Antibodies used were MCL1 (D3CA5, Cell Signaling Technologies), DUX4 (MAB9535, R&D system) and β-actin (NB600-501SS, Novus Biologicals). Western blot analysis of PDX ALL-265 was performed as previously described[68], using the following antibodies: MCL1 (S-19, Santa Cruz Biotechnology) and GAPDH (6C5, Merck Millipore).

**Reporting summary**. Further information on research design is available in the Nature Research Reporting Summary linked to this article.

## Data availability

The RNA-seq data generated in this study have been deposited at the Gene Expression Omnibus under the following accession codes: GSE182760 (*MCL1*), GSE181973 (*MLL-AF4*), GSE182780 (*DUX4-IGH*). Source data are provided with this paper.

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

## Acknowledgements

We thank Liliana Mura, Fabian Klein, Maike Fritschle, Annette Frank and Miriam Krekel for excellent technical assistance; Markus Brielmeier and team (Research Unit Comparative Medicine) for animal care services; Karsten Spiekermann and the LFL laboratory for sequencing the *MLL-AF4* breakpoint; Wolfgang Enard and Helmut Blum for generating SCRB-seq data and Jean Pierre Bourquin and Beat Bornhäuser for providing engrafted sample ALL-265. The work was supported by the Humboldt Postdoctoral Fellowship (to M.C.), and by grants from the European Research Council Consolidator Grant 681524; a Mildred Scheel Professorship by German Cancer Aid; German Research Foundation (D.F.G.); the Collaborative Research Center 1243 "Genetic and Epigenetic Evolution of Hematopoietic Neoplasms", project A05; DFG proposal MA 1876/13-1; Bettina Bräu Stiftung and Dr. Helmut Legerlotz Stiftung (all to I.J.); by the Joint Funding project "Relapsed ALL" of the German Cancer Consortium (DKTK) (to C.B. and I.J.). T.H. was supported by the Physician Scientists Grant (G-509200-004) from the Helmholtz Zentrum München. P.J.J. was supported by the Max Eder-Program grant from the Deutsche Krebshilfe (program #111738), Deutsche José Carreras Leukämie-Stiftung (DJCLS R 12/22 and DJCLS 21 R/2016), Else Kröner Fresenius Stiftung (2014_A185) and Deutsche Forschungsgemeinschaft (DFG FOR 2036, SFB 1335 and SFB 1371).

## Author contributions

M.C. designed and performed experiments, analyzed data and wrote the manuscript, K.V. designed, performed experiments and analyzed data with contributions from J.V. (establishing the technique), M.B. (*DUX4-IGH*), D.S. (data analysis and writing the manuscript), Y.G. (PCR, MCL-1 inhibitor treatment), W.H.L. (cloning), B.V. (establishment of treatment regimens), J.P.S. (quality control experiments); J.W.B. performed and T.H. and V.J. analyzed *DUX4* SCRB-seq data; A.W. and R.M. analyzed *MLL-AF4* RNAseq data; A.A. and V.B. performed zebrafish experiments; V.D. and P.J.J. quantified BH3 protein expression; B.F. and K.R. designed fluorochrome use; C.B., L.B., L.L. and D.M.S. provided PDX models; C.M., M.S.S. and M.B. developed cloning strategies; I.J. supervised the study, and contributed to experimental design, data analysis and writing the manuscript.

## Funding

## Competing interests

P.J.J. has had a consulting or advisory role, received honoraria, research funding, and/or travel/accommodation expenses from: Abbvie, Bayer, Boehringer, Novartis, Pfizer, Servier, BMS and Celgene. The remaining authors declare no competing interests.

## Additional information

[1]Research Unit Apoptosis in Hematopoietic Stem Cells, Helmholtz Zentrum München, German Research Center for Environmental Health (HMGU), Munich, Germany. [2]Laboratory for Leukemia Diagnostics, Department of Medicine III, University Hospital, LMU Munich, Munich, Germany. [3]German Cancer Consortium (DKTK), Partnering Site Munich, Munich, Germany. [4]Department of Pediatrics, Dr. von Hauner Children's Hospital, University Hospital, Ludwig Maximilian University (LMU), Munich, Germany. [5]Clinic and Policlinic for Internal Medicine III, Technical University of Munich, School of Medicine, Munich, Germany. [6]Research Department Cell and Gene Therapy, Department of Stem Cell Transplantation, University Medical Center Hamburg-Eppendorf, Hamburg, Germany. [7]Internal Medicine II, Christian-Albrechts University Kiel and University Medical Center Schleswig-Holstein, Campus Kiel, Kiel, Germany. [8]Department of Pediatrics I, ALL-BFM Study Group, Christian-Albrechts University Kiel and University Medical Center Schleswig-Holstein, Kiel, Germany. [9]Anthropology and Human Genomics, Faculty of Biology, Ludwig Maximilian University (LMU), Munich, Germany. [10]Institute of Pharmaceutical Biology, Diagnosic Center of Acute Leukemias (DCAL), Goethe-University, Frankfurt/Main, Germany. [11]Center for Translational Cancer Research (TranslaTUM), Klinikum rechts der Isar, Technical University of Munich, Munich, Germany. [12]Department of Medicine I, Medical Center, University of Freiburg, Faculty of Medicine, Freiburg, Germany. [13]German Cancer Consortium (DKTK), Partnering Site Freiburg, Freiburg, Germany. [14]Institute of Experimental Hematology, Technical University of Munich, Munich, Germany. [15]These authors contributed equally: Michela Carlet, Kerstin Völse. ✉email: Irmela.Jeremias@helmholtz-muenchen.de

