## [Peer Review File · Nature Communications]

In vivo inducible reverse genetics in patients' tumors to identify individual therapeutic targetsREVIEWER COMMENTS

Reviewer #2 (Remarks to the Author); expert on genetics and leukaemia

Carlet et al present a viral system for inducible gene knockdown in patient-derived xenografts. The authors use sequential rounds of fluorescently labeled viral vectors to deliver a CreER-GLuc construct and a Cre inducible shRNA to allow for Tamoxifen inducible repression of genes of interest. Orthogonal fluorescent proteins allow for clear demarcation of cells harboring control shRNAs as well as ones targeted to genes of interest, providing internal controls for experiments of interest. The authors proceed to utilize shRNA targeting fusion proteins of interest to demonstrate the specificity and fidelity of interest. While the specificity of the authors approach is carefully controlled, there is not much in the way of new insights offered in the experiments described here. While individually the DUX4 and MCL1 dependencies are of general interest, there is a lack of followup/ discovery studies beyond proof of concept examples that the system described in Figure 1 works as intended. However, the FLEX switch Cre inversion model (originally FLIP, from Stern et al 2008, PNAS), is a well-established method in the field for inducible shRNA control, and has been applied in vivo in both the original article as well many times thereafter. As noted by the authors, one adaptation of this system is the temporal induction of CreER activity with the pulse of Tamoxifen, allowing for in vivo mechanistic studies. Indeed, it would be of general interest to test knockdown of oncogenes late or early in disease progression, however no such experiments were described here. The lack of new biological/mechanistic insights into oncogene or therapeutic vulnerability biology leave this story underdeveloped. Additional points: 1. It is unclear the utility of performing secondary transplants into Zebrafish, especially given the usage of the GLuc and ability to track tumor cells in mice with bioluminescent imaging. 2. The authors should remove reference to a Gene of Interest in Figure 1 and replace it with the actual gene that was targeted by the shRNA. Example data is insufficient. 3. The data presentation in Figure 2G does not accurately reflect the legend. Either error bars are not visible due to very low standard error of measure, in which case the authors should plot individual data points, or only a single data point is provided. 4. Figure 2D and Figure 2I appear to indicate that the shCTRL/shCTRL mixtures were not treated with TAM. Why this data point is included on the graph is currently unclear. Were control TAM treatments on this group performed? Is the reader to assess differences at day 3 between the shCTRL and shMCL1 or shDUX4 group? 5. Y axis or legend for Figure 2J should provide a more descriptive account of what is meant by "mRNA level". 6. No author contribution provided for LL or LB. 7. For Supplemental Figure 2J, full image of uncropped western films should be provided in the supplemental data.

Reviewer #3 (Remarks to the Author); expert on leukaemia and single-cell analysis

As part of a future precision medicine program, each patient would receive a small molecule inhibitor based on the genetics of their cancer. However, to date, it has been difficult to ascertain which particular mutations, vulnerabilities to exploit. In the paper by Carlet et al, the authors design an elegant in vivo PDX based model system to knockdown genes that may be potential therapeutic targets. The knockdown rather than knockout is used to phenocopy a potential small molecule-based intervention where some residual activity might be predicted to exist. Elegantly, the authors also include an internal non-targeting shRNA mir control in each mouse to help validate the knockdown - which is mixed at a 1:1.

The authors then test their model targeting MCL1, DUX4 and an MLL-AF4.

The paper is well written and clear and the data is presented in a nice coherent manner.

Major point:

1) Although not explicitly mentioned, if these models were to be used prospectively for a patient in real-time, the timeline for the establishment of these models would be good to know. There are some methodological details stating the cells were cultured for four days, sorted, and then injected. How long did it take to generate each PDX model (including all the transduction steps etc). Can the authors provide details on the transduction efficiency for each of the PDX's and the time to engraftment? Was the same efficiencies seen for the Cre-ER vectors and the shRNA?

2) Have the authors compared the genetics of the PDX after the two transductions (e.g. the CreER-T2 and the knockdown vectors) to show that there has not been any clonal selection compared to the original sample? This would be relevant for those PDX cells that showed a very low transduction efficiency.

3) It is not entirely clear what the criteria are for starting TAM administration (e.g. the level of leukemia burden). Is this done solely on luminescence? How is the 1:1 ratio checked that there is a change in engraftment (leakiness?) because although the majority of ALL PDX's start at 50% - the AML-491 and AML-393 after 3 days TAM (Fig S2 C and D) are already at 20%? This is not the case for AML-388 however in Figure 2D which shows 50% Can the authors provide details and provide evidence that the AML-PDX were at 1:1 ratio to begin with?

4) Further to point 3 above- the MCL1 KD data supporting a vulnerability in the AML PDXs are overstated. The engraftment 3 days after TAM treatment is already at ~20%. The authors claim a statistical difference - although the biological relevance of what appears to be as little as a 3-5% decrease is questionable? The statistical test for what appears to be only n=2 data points in Figure S2D is odd given the std deviation bar yet still highly significant? Can the authors elaborate on the appropriateness of the statistical test used.

5) The authors have stated that the KD of MCL1 decreasing growth validates it as a therapeutic vulnerability. The authors should now test this statement using the MCL1 inhibitors now available as a single agent. This should be tested ex vivo in a dose-response curve analysis and also in vivo using the PDX's to answer whether the predicted PDX's equally sensitive to the MCL 1 inhibitors? Then the authors might then consider revising statements on whether MCL1 is a true vulnerability .(E.g. targeting MCL1 alone is insufficient for therapeutic effect.)

6) On line 128-129 the manuscript reference Figures S2c-e) showing apoptosis but this is limited to Annexin V flow cytometry (rather than classical Annexin V/PI). The authors should provide additional data to support this statement that apoptosis is occurring (e.g. cleaved Caspase). This should also be done for AML-491 and 393 and the ALL PDX to see if the genetic KD phenocopies the small molecule inhibition.

Minor Points

1) In figure 1C, Line 87, the authors state that targeting an essential exemplary gene reveals this population has a competitive disadvantage. What gene did the authors target for this figure? It is not explicitly stated although the legend for Figure S1 says this is MCL1? I would not categorize MCL1 as an exemplary essential gene and so perhaps the authors could provide some further clarity in the text in what they refer to as an "exemplary essential gene".

2) Line 103-104. Consider revising the sentence - the data shows that the shRNA does not lead to down regulation of MLL or AF4 - but the sentence says "it might induce minor adverse effects on normal tissue....". I think you mean that there will be little effect on normal tissues

3) Figure 2A - the qPCR reveals that the shRNA has no effect on the MLL or AF4 in patient ALL-707. It is not clear from the "relative" value the level of expression of MLL and AF4 is (e.g. the wild type alleles that were not translocated). Did these have a similar Ct value as the fusion? Or were they expressed as very low levels because the fusion alleles are expressed very high? TO this end, did the authors check in a patient/cell line that did not have this fusion but do express higher levels of MLL or

AF4 to show no off-target effects

4) Line 48-49 - The authors state that a single integration per cell genome was achieved. In the experience of this reviewer, the cut off for such a statement is that the efficiency of transduction should be below 30%. Yet this data presented showed mCHERRY to be at 97% - albeit this is likely to be after the sorting. You can only claim single genome integration events using a genetic-based assay - both for the Cre-ER but more importantly for the shRNAmir constructs. This again reflects the lack of details for the transduction efficiencies achieved mentioned above. The authors, therefore, need to undertake a more robust analysis for this claim or revise the statement.

Reviewer #4 (Remarks to the Author); expert on leukaemia, familiar with zebrafish models

The authors have developed an inducible system to indentify critical target genes in acute leukemias. The system is elegant and elaborate and the manuscripts focuses on proving the workings of the experimental system, which is shown in case of MLL-AF4 (knockdown of the fusion) and DUX4-rearranged ALL (knockdown of MCL1). Although this work already is technically very sound and of high level, I would kindly suggest the authors to consider taking the next step, and perform an unbiased screen for target genes by using the current system and choosing a single AL sub-entity/sub-type (e.g. DUX4-re ALL), and then conducting a screen for potential target genes that the leukemia cells are dependent on.

I have some additional more technical comments:

1) MLL-AF4 knockdown by mRNA-level is approximately 40%, and yet the leukemia cells seem to be in decline. Is the diminishing counts due to even decrease of fusion gene, ie. are there cells with 100% KO and others with yet full expression? Furthermore, it would be better suited to show a Western blot of MLL-AF4 knockdown (if working antibodies are available), instead or in addition to RT-qPCR.

2) In Figure 2a (and in others when possible) the actual data points should be shown (instead of mean/SEM), e.g. by scatter plot.

3) Maybe it would be better to show the apoptosis assay (e.g. Annexin V/PI) itself, instead of gene expression signature in Fig 2f.

4) Fig 2h would benefit from having a single treatment of cells with venetoclax only in order to isolate the effect of venetoclax to the cells. And maybe then showing combination index (CI) to prove whether they (anti-BCL2 drug + MCL1 KD) are additive or synergistic when working together.

5) The authors are showing altered expression of three selected genes in DUX4-knockdown cells yet discuss about "transcriptomic" analysis. The DUX4/ERG subtype carries a typical transcriptomic signature, and therefore I suggest that the authors either show the whole "DUX4/ERG-signature" (before and after knockdown) or prove the downstream effect otherwise (e.g. lack of ERGalt expression). The downregulation fo DUX4 gene itself by mRNA /Western blot should be also shown in the main figure, similar to MCL1.

6) The selection of MCL1 gene as one of the "genes of interest" is unclear to me. Maybe this should be elaborated a bit more. Or was this the best one of the tested genes that worked out?

7) In the zebrafish assay, only single time point is shown. The selection of a single time point (why not a serial assessment) and its timing should be clarified along with the zf line used (wt or immunocompromised), the age of larvae at the time of assessment, and the ethical aspects related to animal welfare as well (need for permits?). I am also wondering what "extra" this assay provides to the readers...

Point by Point response to reviewer comments

Carlet et al., NCOMMS-20-24554-T

*In vivo inducible reverse genetics in patients' tumors
to identify individual therapeutic targets*

We are happy to see that our manuscript was well received and thank all reviewers for their scientifically sound comments.

Motivated by the constructive points raised, we extensively revised the manuscript, generated new transgenic AML models, performed new *in vivo* treatment trials, included data from 130 additional mice and addressed all points raised. The large amount of new data resulted in extending previously 2 to now 3 printed Figures and adding well above 20 new panels to supplemental Figures, together with 2 new supplemental tables, with contributions from 7 additional co-authors.

Of major importance for precision oncology, our novel technology enables validating vulnerabilities at the individual patient level *in vivo*, which was impossible until now. We study patient-derived tumors and induce knockdown in pre-established leukemias *in vivo*, which faithfully mimics the situation of treatment in patients, characterized by existence of established tumors.

New data include, but are not restricted to

- the novel biologic insight that DDIT4L represents a yet unknown downstream target of DUX4 with essential function in ALL (new Figure 3e-i and S3e-h)
- new PDX *in vivo* treatment trials showing that sensitivity to treatment with an MCL-1 inhibitor correlates to results from our novel inducible knockdown technique (new Figure 2d-f and S2h-i)
- new PDX *in vivo* trials showing that the effect of inducible knockdown was independent from tumor load (new Figure S2a)
- detailed data on generation of transgenic PDX models (new Table S1 and S2)
- new descriptive data including
 - gene expression profile upon knockdown of DUX4 (new Figure S3e-g)
 - clinical patient data (new Table S1)

Detailed responses to each individual point raised by the reviewer are found below, with the points highlighted by the editor marked in bold.

REVIEWER COMMENTS

Reviewer #2; expert on genetics and leukaemia

Carlet et al present a viral system for inducible gene knockdown in patient-derived xenografts. The authors use sequential rounds of fluorescently labeled viral vectors to

deliver a CreER-GLuc construct and a Cre inducible shRNA to allow for Tamoxifen inducible repression of genes of interest. Orthogonal fluorescent proteins allow for clear demarcation of cells harbouring control shRNAs as well as ones targeted to genes of interest, providing internal controls for experiments of interest. The authors proceed to utilize shRNA targeting fusion proteins of interest to demonstrate the specificity and fidelity of interest.

While the specificity of the authors approach is carefully controlled, there is not much in the way of new insights offered in the experiments described here. While individually the DUX4 and MCL1 dependencies are of general interest, there is a lack of follow up/discovery studies beyond proof of concept examples that the system described in Figure 1 works as intended. However, the FLEX switch Cre inversion model (originally FLIP, from Stern et al 2008, PNAS), is a well-established method in the field for inducible shRNA control, and has been applied *in vivo* in both the original article as well many times thereafter. As noted by the authors, one adaptation of this system is the temporal induction of CreER activity with the pulse of Tamoxifen, allowing for *in vivo* mechanistic studies.

We thank the reviewer for the thorough evaluation and most helpful comments. As suggested, we performed new experiments and discovered the biological novelty that the DUX4-regulated gene DDIT4L represents a novel essentiality in ALL (new Figure 3e-i and S3e-h; see general point 2 for details).

The Cre/FLIP inducible knockdown system has never been used in PDX models *in vivo* before. As PDX models mimic the clinical situation of patients more closely than GEMMs or cell lines, our manuscript represents an important technical advance and genuine innovation. Due to major challenges, only a minimal number of studies demonstrated efficient manipulation of gene expression in PDX models *in vivo* (Hulton et al., *Nature Cancer* 2020; Miller et al., *Nature* 2017), entirely lacking inducible knockdown approaches up to now.

General points

1. Indeed, it would be of general interest to test knockdown of oncogenes late or early in disease progression, however no such experiments were described here.

We performed new experiments and now demonstrate that knockdown of MCL1 had highly similar effects, late and early in disease progression (new Figure S2a). The new data confirm that our novel technique is able to measure gene essentialities at different disease stages of PDX models *in vivo*.

2. **The lack of new biological/mechanistic insights into oncogene or therapeutic vulnerability biology leave this story underdeveloped.**

We performed new experiments and followed up on the gene expression profile upon DUX4 knockdown in PDX cells *in vivo*. We asked whether any of the downregulated genes would play an essential function itself so that downregulation would alter leukemia growth.

Little is known of the downstream target DDIT4L, which has been shown to regulate mTOR signalling and autophagy in mammalian cells (Corradetti et al., *JBC* 2005; Miyazaki and Esser, *Am J Physiol Cell Physiol* 2009; Simonson et al., *Sci Signal* 2017). DDIT4L expression is induced in the presence of different types of pathological stress, suggesting a possible involvement of DDIT4L in stress response.

We found DDIT4L highly expressed in DUX4 rearranged ALL (new Figure S3h). Using our inducible knockdown *in vivo* approach, we now

discovered that DDIT4L represents a novel therapeutic vulnerability, demonstrating that the DUX4-regulated gene DDIT4L is essential in ALL, in pre-established leukemias *in vivo* (new Figure 3e-i and S3e-h). We conclude that DDIT4L might represent a therapeutic target in DUX rearranged ALL.

Additional points:

1. It is unclear the utility of performing secondary transplants into Zebrafish, especially given the usage of the GLuc and ability to track tumor cells in mice with bioluminescent imaging.

We introduced the Zebrafish model (i) to quality control our inducible knockdown approach by a second independent *in vivo* model system; (ii) to add microscopy as further independent readout, complementing GLuc-based *in vivo* imaging and fluorochrome-based flow cytometry; (iii) to gain insights into localization of individual cells in the *in vivo* niche; and (iv) to analyse the subpopulations in the same animals at different time points.

During the revision and stimulated by the suggestion of reviewer 4, point 7, we performed new experiments to increase numbers of animals, time points and improve statistics (improved Figure S2f).

At the reviewers' discretion, we offer removing Zebrafish data.

2. The authors should remove reference to a Gene of Interest in Figure 1 and replace it with the actual gene that was targeted by the shRNA. Example data is insufficient.

We removed the term "exemplary gene" and now clearly indicate that MCL1 was studied, in new Legend to Figure 1C. Page 5 now indicates that "For exemplary purposes and to describe distinct aspects of the method, the apoptosis regulator MCL1 was chosen as GOI (Figures 1 and S1)"

3. The data presentation in Figure 2G does not accurately reflect the legend. Either error bars are not visible due to very low standard error of measure, in which case the authors should plot individual data points, or only a single data point is provided.

As pandemic conditions did unfortunately not allow performing high numbers of *in vivo* trials, we removed data on AML-393 and AML-491 and thus previous Figure 2G from the manuscript; instead, we directed the restricted number of *in vivo* experiments towards the new target DDIT4L.

4. Figure 2D and Figure 2I appear to indicate that the shCTRL/shCTRL mixtures were not treated with TAM. Why this data point is included on the graph is currently unclear. Were control TAM treatments on this group performed? Is the reader to assess differences at day 3 between the shCTRL and shMCL1 or shDUX4 group?

All animals were treated with TAM, including shCTRL/shCTRL animals, except the few control animals which received the solvent (Figure S1f); this is now clearly stated in all legends. Statistical analysis compares the percentage of cells with knockdown of the gene of interest between starting conditions (3 days after TAM) to later time points. To avoid misunderstandings, we moved all shCTRL/shCTRL data to the supplement (new Figure S1d).

5. Y axis or legend for Figure 2J should provide a more descriptive account of what is meant by “mRNA level”.
Labelling of the Y-axis was changed to “normalized mRNA expression (log2)” in revised Figure 3g.
6. No author contribution provided for LL or LB.
We added “LL and LB provided PDX models.”
7. For Supplemental Figure 2J, full image of uncropped western films should be provided in the supplemental data.
We added raw data of all protein expression analyses by capillary immunoassay or conventional Western Blots at the end of “Supplementary information” file.

Reviewer #3: expert on leukaemia and single-cell analysis

As part of a future precision medicine program, each patient would receive a small molecule inhibitor based on the genetics of their cancer. However, to date, it has been difficult to ascertain which particular mutations, vulnerabilities to exploit. In the paper by Carlet et al, the authors design an elegant *in vivo* PDX based model system to knockdown genes that may be potential therapeutic targets. The knockdown rather than knockout is used to phenocopy a potential small molecule-based intervention where some residual activity might be predicted to exist. Elegantly, the authors also include an internal non-targeting shRNAmir control in each mouse to help validate the knockdown - which is mixed at a 1:1.

The authors then test their model targeting MCL1, DUX4 and an MLL-AF4.

The paper is well written and clear and the data is presented in a nice coherent manner.

We deeply thank the reviewer for appreciating our work.

Major points:

1. Although not explicitly mentioned, if these models were to be used prospectively for a patient in real-time, the timeline for the establishment of these models would be good to know. There are some methodological details stating the cells were cultured for four days, sorted, and then injected. How long did it take to generate each PDX model (including all the transduction steps etc)? Can the authors provide details on the transduction efficiency for each of the PDX's and the time to engraftment? Was the same efficiencies seen for the Cre-ER vectors and the shRNA?

The desired information was added into new Tables S1 and S2. In brief, *in vivo* trials were performed by earliest in passage 3 as several passages were needed for transducing and enriching double transgenic PDX cells. A passage typically took 6-12 weeks; transduction efficiencies typically ranged from 1-30% (Table S2), which should lead to bona fide single integrations per genome, as also discussed in minor point 4; transduced cells were enriched by flow cytometry gating on the transgenic fluorochrome. As Cre-ER^{T2} is larger in size, transduction efficiencies were indeed lower compared to the knockdown FLIP construct.

2. Have the authors compared the genetics of the PDX after the two transductions (e.g. the CreER-T2 and the knockdown vectors) to show that there has not been any clonal selection compared to the original sample? This would be relevant for those PDX cells that showed a very low transduction efficiency.

A major advantage of our inducible approach lies in the fact that it perfectly controls for putative clonal bias, as it compares the same cells with and without induction of knockdown. In addition, we quality controlled experiments by (i) mixing independently transduced shCTRL cells to shGOI cells in the same mouse and (ii) treating animals with solvent (new Figure S1d and Figure S1e-f).

On a more general level, we extensively quality controlled lentiviral transduction of PDX leukemia models in a previous publication (Vick et al., Plos One 2015). In brief, the parameters studied (clonal composition according to AML-specific mutations, proliferation rate, drug sensitivity, surface marker expression) did not differ substantially between untransduced and transduced PDX models, even upon low lentiviral transduction efficiencies. As putative explanation, PDX models are transduced which have putatively gained a clonal equilibrium. For the reviewer's information, a table on AML-specific mutations of PDX samples before and after lentiviral transduction and different transduction efficiencies is attached at the end of this document, complementing and exceeding our previously published data.

We added to page 6

“(In) our previous studies (Vick et al., 2015), we found that transduction and enrichment of PDX cells was not associated with clonal selection, and that PDX samples largely maintained their sample-specific mutational pattern.”

3. a) It is not entirely clear what the criteria are for starting TAM administration (e.g. the level of leukemia burden). Is this done solely on luminescence?

We performed new experiments and now show that application of TAM leads to identical induction of fluorochrome switch (Figure S1c) and to identical growth disadvantage (new Figure S2a), independent of tumor burden.

While TAM can be administered at any time during tumor growth, we preferred early time points, typically 7 days after transplantation; this is now clearly stated in all Figures. Early application of TAM allows for longer observation periods upon knockdown, before mice succumb to death by leukemia due to shCTRL cells.

- b) How is the 1:1 ratio checked that there is a change in engraftment (leakiness?) because although the majority of ALL PDX's start at 50% - the AML-491 and AML-393 after 3 days TAM (Fig S2 C and D) are already at 20%? This is not the case for AML-388 however in Figure 2D which shows 50%. Can the authors provide details and provide evidence that the AML-PDX were at 1:1 ratio to begin with?

We took this point very seriously and improved data analysis and presentation. New statistical analysis is now exclusively restricted to the shGOI/shCTRL mixture over time; it now compares the ratio of both populations between starting conditions (3 days after TAM) and later time points. As major advantage, any putative deviation in the initial mixture is always controlled for.

Accordingly, data on the shCTRL/shCTRL mixture were moved to the supplement (new Figure S1d). All experiments throughout the paper were re-analysed using the improved statistical approach - which yielded highly similar results to the previous way of analysis.

In addition and stimulated by the reviewer's comments, we now performed more detailed new experiments for MCL1. We indeed found that MCL1 knockdown induced a very rapid anti-tumor effect, starting 54 hours after TAM in AML-388 (Figure S2c) and in AML-491 (see Figure below for review only), indicating that deviation from the 1:1 mixture at d3 post-TAM might be a sign of rapid cell death induction due to GOI loss. As consequence and as pandemic conditions did unfortunately not allow repeating all experiments, we removed data on AML-393 and AML-491 from the manuscript.

4. a) Further to point 3 above - the MCL1 KD data supporting a vulnerability in the AML PDXs are overstated. The engraftment 3 days after TAM treatment is already at ~20%. The authors claim a statistical difference - although the biological relevance of what appears to be as little as a 3-5% decrease is questionable?

We agree to the reviewer's comment; as we were unfortunately unable to repeat all experiments due to pandemic conditions, we decided removing data on AML-393 and AML-491. Instead, we concentrated on DDI4TL for adding a biologic novelty to our manuscript.

- b) The statistical test for what appears to be only n=2 data points in Figure S2D is odd given the std deviation bar yet still highly significant? Can the authors elaborate on the appropriateness of the statistical test used.

We performed new experiments and increased mouse numbers to at least 3 animals at each time point in each *in vivo* experiment in the revised version.

Statistical analysis is now performed by comparing the percentage of cells with knockdown of the gene of interest between starting conditions (3 days after TAM) to later time points, for which student t-test is the appropriate statistical test.

5. The authors have stated that the KD of MCL1 decreasing growth validates it as a therapeutic vulnerability. The authors should now test this statement using the MCL1 inhibitors now available as a single agent. This should be tested *ex vivo* in a dose-response curve analysis and also *in vivo* using the PDX's to answer whether the predicted PDX's equally sensitive to the MCL 1

inhibitors? Then the authors might then consider revising statements on whether MCL1 is a true vulnerability. (E.g. targeting MCL1 alone is insufficient for therapeutic effect.)

We performed new experiments and tested the MCL-1 inhibitor S63845 as single agent *in vivo*. New Figure 2d-f indicates that AML-388, but not ALL-199 shows sensitivity towards treatment with the MCL-1 inhibitor. Thus, MCL-1 represents a true vulnerability for AML-388, but not ALL-199 which is now clearly stated in the text. Molecular data correlated with data from the *in vivo* treatment trial, highlighting the usefulness of our novel technique.

6. On line 128-129 the manuscript reference Figures S2c-e) showing apoptosis, but this is limited to Annexin V flow cytometry (rather than classical Annexin V/PI). The authors should provide additional data to support this statement that apoptosis is occurring (e.g. cleaved Caspase). This should also be done for AML-491 and 393 and the ALL PDX to see if the genetic KD phenocopies the small molecule inhibition.

Unfortunately, it is technically unfeasible to perform AnnexinV/PI staining in the presence of the 5 fluorophores required for the competitive inducible knockdown system and too few cells could be re-isolated from mice to perform Caspase Western Blot. To nevertheless strengthen that apoptosis is the form of cell death occurring upon *mcl-1* knockdown, we analysed AnnexinV in additional samples (improved Figure S2e) and performed GSEA analysis, where we find HALLMARK_Apoptosis genes enriched in the 72h versus 24h time point in the shMCL1 population (Figure S2d).

Minor Points

1. In figure 1C, Line 87, the authors state that targeting an essential exemplary gene reveals this population has a competitive disadvantage. What gene did the authors target for this figure? It is not explicitly stated although the legend for Figure S1 says this is MCL1? I would not categorize MCL1 as an exemplary essential gene and so perhaps the authors could provide some further clarity in the text in what they refer to as an “exemplary essential gene”.

We removed the term “exemplary gene” and now clearly indicate that MCL1 was studied, in new Legend to Figure 1C. Page 5 now indicates that

“For exemplary purposes and to describe distinct aspects of the method, the apoptosis regulator MCL1 was chosen as GOI (Figures 1 and S1)”

2. Line 103-104. Consider revising the sentence - the data shows that the shRNA does not lead to down regulation of MLL or AF4 - but the sentence says “it might induce minor adverse effects on normal tissue....”. I think you mean that there will be little effect on normal tissues

The sentence was revised to

“...as the shRNA sequence targeted neither of the individual wildtype genes, MLL or AF4 (Figure 3a and S3b), no major adverse effects on normal tissue are expected when applied in vivo...”

3. Figure 2A - the qPCR reveals that the shRNA has no effect on the MLL or AF4 in patient ALL-707. It is not clear from the “relative” value the level of expression of MLL and AF4 is (e.g. the wild type alleles that were not translocated). Did these have a similar Ct value as the fusion? Or were they expressed as very low levels because the fusion alleles are expressed very high? To this end, did the authors check in a patient/cell line that did not have this fusion but do express higher levels of MLL or AF4 to show no off-target effects?

We added data on qPCR analysis of the translocation-negative ALL-265 PDX sample as new Figure S3b; we did not see significant reduction of either wildtype MLL or AF4 upon expressing the shRNA against the MLL-AF4 fusion. Figure for review only below shows basic Ct values for PDX ALL-707 with MLL-AF4 rearrangement (MLLr) and PDX ALL-265 without MLLr. While mRNA for the MLL-AF4 fusion was selective for ALL-707, expression levels of MLL were similar between both samples and similar to MLL-AF4 level in ALL-707, while AF4 was higher expressed in MLLr positive ALL-707.

4. Line 48-49 - The authors state that a single integration per cell genome was achieved. In the experience of this reviewer, the cut off for such a statement is that the efficiency of transduction should be below 30%. Yet this data presented showed mCHERRY to be at 97% - albeit this is likely to be after the sorting. You can only claim single genome integration events using a genetic-based assay - both for the Cre-ER but more importantly for the shRNAmir constructs. This again reflects the lack of details for the transduction efficiencies achieved mentioned above. The authors, therefore, need to undertake a more robust analysis for this claim or revise the statement.

New Table S2 now indicates that transduction efficiencies were typically well below 30%; methods now highlights that transduced cells were enriched by flow cytometry. We toned down our statement on page 4 and now write:

“Transduction efficiencies were typically well below 30% (Table S2), putatively indicating a single viral integration per genome according to literature (Charrier et al., Gene Therapy 2011)...”

Reviewer #4: expert on leukaemia, familiar with zebrafish models

The authors have developed an inducible system to identify critical target genes in acute leukemias. The system is elegant and elaborate and the manuscripts focuses on proving the workings of the experimental system, which is shown in case of MLL-

AF4 (knockdown of the fusion) and DUX4-rearranged ALL (knockdown of MCL1). Although this work already is technically very sound and of high level, I would kindly suggest the authors to consider **taking the next step**, and perform an unbiased screen for target genes by using the current system and **choosing a single AL sub-entity/sub-type (e.g. DUX4-re ALL)**, and then conducting a screen for **potential target genes that the leukemia cells are dependent on**.

We thank the reviewer for the thorough evaluation and for acknowledging our work.

While we agree that performing unbiased screens in PDX models *in vivo* is a highly attractive and relevant goal, establishing such *in vivo* screens in PDX models is very challenging. Literature mostly reports on constitutive screens using cell lines *in vitro*, contrasting our current inducible PDX *in vivo* work. In a project independent from the present one, we take more than 2 years meanwhile for establishing *in vivo* screens in PDX leukemias, this time using CRISPR-Cas9 and a constitutive instead of an inducible approach – and thus unrelated from the present project.

We nevertheless took a next step, albeit in a slightly different approach. We performed new experiments to study DUX4 downstream target genes. Little is known of DDIT4L which has been shown to regulate mTOR signalling and autophagy in mammalian cells (*Corradetti et al., JBC 2005; Miyazaki and Esser, Am J Physiol Cell Physiol 2009; Simonson et al., Sci Signal 2017*). Using our inducible knockdown *in vivo* approach, we now describe for the first time that the DUX4-regulated gene DDIT4L represents an essentiality in ALL, in pre-established leukemias *in vivo* (new Figure 3e-i and S3e-g). We conclude that DDIT4L might represent a therapeutic target in DUX rearranged ALL.

I have some additional more **technical comments**:

1. MLL-AF4 knockdown by mRNA-level is approximately 40%, and yet the leukemia cells seem to be in decline. Is the diminishing counts due to even decrease of fusion gene, ie. are there cells with 100% KO and others with yet full expression? Furthermore, it would be better suited to show a Western blot of MLL-AF4 knockdown (if working antibodies are available), instead or in addition to RT-qPCR.

We performed new experiments and tried to measure expression levels of the fusion mRNA in single cells, but our attempt was unfortunately unsuccessful. We tried to perform single cell qPCR but experienced technical challenges, and more routine RNA seq assays like the 10x genomics Chromium platform were not suitable to discriminate between the wildtype and fusion gene.

Regarding Western Blotting of the MLL-AF4 fusion protein, we newly collaborated with the Marschalek group which are experts on biology of MLL rearrangements; they recommended relying on transcriptome analysis which we included as Figures S3c-d, as Western Blot is, at this time, technically unfeasible.

2. In Figure 2a (and in others when possible) the actual data points should be shown (instead of mean/SEM), e.g. by scatter plot.

We now show scatter plots in all subpanels, including mentioned Figure 3a.

3. **Maybe it would be better to show the apoptosis assay (e.g. Annexin V/PI) itself, instead of gene expression signature in Fig 2f.**

We performed new experiments and now show Annexin V staining for all samples in revised Figure S2e; unfortunately, a combined AnnexinV/PI staining is unfeasible in the presence of the 5 fluorochromes used.

4. Fig 2h would benefit from having a single treatment of cells with venetoclax only in order to isolate the effect of venetoclax to the cells. And maybe then showing combination index (CI) to prove whether they (anti-BCL2 drug + MCL1 KD) are additive or synergistic when working together.

Revised Figure 2g now shows that Venetoclax alone had no effect in absence of MCL1 knockdown, but only in presence of MCL1 knockdown.

For calculating synergism, the combination index (CI) was unsuitable in our trial, as we used a single dose of Venetoclax. Instead, we used the fractional product method (*Webb JL. Effect of more than one inhibitor. In: Hochster RM, Quastel JH, eds. Enzyme and Metabolic Inhibitors. Vol 1. New York, NY: Academic Press; 1963:487-512.*) which indicated that combining MCL1 knockdown with Venetoclax induced synergistic anti-leukemia effects. We introduced the following paragraph into the Online Method section.

Synergistic effect was calculated using the fractional product method: Measured survival rates were 0.39 upon MCL1 KD and 1.0 upon Venetoclax; expected apoptosis induction of independent application of Mcl-1 knockdown and Venetoclax was calculated as [(1 minus (survival after simulation with Mcl-1 knockdown) times (survival after stimulation with VCR)) times 100] which resulted to be 0.61; measured apoptosis by the combination of MCL1 and Venetoclax was 0.94 and thus much higher than the expected apoptosis of 0.61, proving that the combination acted in a synergistic way.

5. **The authors are showing altered expression of three selected genes in DUX4-knockdown cells yet discuss about "transcriptomic" analysis. The DUX4/ERG subtype carries a typical transcriptomic signature, and therefore I suggest that the authors either show the whole "DUX4/ERG-signature" (before and after knockdown) or prove the downstream effect otherwise (e.g. lack of ERGalt expression). The downregulation of DUX4 gene itself by mRNA /Western blot should be also shown in the main figure, similar to MCL1.**

We now display the entire DUX4-signature before and after DUX4 knockdown (new Figure 3e). Indeed, the new analysis suggested by the reviewer revealed highly helpful; signs of the ERGalt DUX4-IGH transcriptomic signatures (*Harvey et new data analysis al., Blood 2010; Tanaka et al., Haematologica 2018*) were decreased upon DUX4 knockdown (Figure 3e-f, S3e-g).

We now display downregulation of DUX4 on protein level in main Figure 3d.

6. The selection of MCL1 gene as one of the "genes of interest" is unclear to me. Maybe this should be elaborated a bit more. Or was this the best one of the tested genes that worked out?

MCL1 was selected from literature, not from a gene list. MCL-1 represents an attractive proof of principle target gene, as it is intensively discussed as therapeutic target in acute leukemia, but literature suggests that not all patients respond to MCL1 treatment (*Khaw et al., Blood 2020; Kotschy et al., Nature*

2016). Thus, MCL-1 was likely to display an essential function in some, but not all PDX models which allowed validating that our method discriminates between responsive and non-response PDX models. This is now clarified in the text, page 7:

“We selected MCL-1 as proof of principle target gene from literature as certain, but not all leukemias seem responsive to MCL1 inhibition^{30,31}.”

7. In the zebrafish assay, only single time point is shown. The selection of a single time point (why not a serial assessment) and its timing should be clarified along with the zf line used (wt or immunocompromised), the age of larvae at the time of assessment, and the ethical aspects related to animal welfare as well (need for permits?). I am also wondering what "extra" this assay provides to the readers...

We performed new experiments and now show a serial assessment with clearly indicated time points (new Figure S2f); monitoring time points later than 3 days after fertilization was not reliable due to an increase of unspecific signal (increased debris of dead cells and fish auto fluorescence).

We now clearly indicate that we used embryos/larvae for less than 5 days after fertilization which does not require immunosuppression; we used wild type zebrafish (AB line); larvae were transplanted 2 days after fertilization. We added into the Online Method section:

“Zebrafish embryos/larvae were studied exclusively within the first 5 days after fertilization, handled compliant to local animal welfare regulations and maintained according to standard protocols (www.ZFIN.org) which does not require a special permit according to German Laboratory Animal Protection Law.”

We introduced the Zebrafish model (i) to quality control our inducible knockdown approach by a second independent in vivo model system; (ii) to add microscopy as further independent readout, complementing GLuc-based in vivo imaging and fluorochrome-based flow cytometry; (iii) to gain insights into localization of individual cells in the in vivo niche; and (iv) to analyse the subpopulations in the same animal at different time points. At the reviewers' discretion, we offer removing Zebrafish data.

Data for Reviewer:

Table: Clonal composition of PDX following lentiviral transduction

AML PDX sample	PDX mutations	VAF PDX	TE*	VAF t-PDX
AML-388	CEBPZ:NM_005760:exon16:c.3143_3146AAA; frameshift substitution	0.51	2.3%	ND [#]
	KRAS:NM_004985:exon3:c.183A>C:p.Q61H; nonsynonymous SNV	0.34		0.42
AML-393	KRAS:NM_033360:exon2:c.G35C>p.G12A; nonsynonymous SNV	0.47	12.5%	0.46
	BCOR:NM_017745:exon5:c.3035_3038del;p.1012_1013del; frameshift deletion	0.42		0.46
AML-491	ETV6:NM_001987:exon5:c.641C>T:p.P214L; nonsynonymous SNV	0,42	11%	0,44
	DNMT3A:NM_175629:exon23:c.2644C>A:p.R882S; nonsynonymous SNV	0,37		0,56
	RUNX1:NM_001754:exon5:c.408T>G:p.N136K; nonsynonymous SNV	0,54		0,55
	BCOR:NM_017745:exon4:c.2048_2049C; frameshift substitution	0,55		0,48
	JAK1:NM_002227:exon14:c.1972G>T:p.V658F	0,00		0,01
	PTPN11:NM_002834:exon3:c.181G>C:p.D61H; nonsynonymous SNV	0,49		0,43
	NRAS:NM_002524:exon3:c.181C>A:p.Q61K	0,08		0,01
	KRAS:NM_004985:exon2:c.35G>C:p.G12A,	0,36		0,49
	EZH2:NM_004456:exon18:c.2075C>G:p.A692G (as in 661) (FISH: 7q31-Deletion (45/120))	0,00		0,00
AML-415	IDH1:NM_005896:exon4:c.395G>A:p.R132H; nonsynonymous SNV	0,52	22%	0,52
	FLT3:NM_004119:exon14:c.1793_1794ins(TGATTTTCAGAG AATATGA);p.E598delinsDDFREYE; nonframeshift insertion	0,84		1,00
	DNMT3A:NM_153759:exon14:c.1579G>A:p.V527I; nonsynonymous SNV	0,65		0,51
	DNMT3A:NM_175629.2:c.2141C>G, p.(Ser714Cys)	0,34		0,48
	NPM1:NM_002520:exon11:c.859_860insTCTG;p.L287; frameshift insertion	0,51		0,52
AML-573	DNMT3A:NM_022552:exon17:c.1988C>T:p.S663L,	0,41	2,5%	0,35
	DNMT3A:NM_022552:exon16:c.1930G>T:p.A644S,	0,46		0,44
	FLT3:NM_004119:exon14:c.1811_1812insTCCCTCAGATAA TGAGTACTTCTACGTTGATTTTCAGAGAATATGAATATGAT CTCAAATGGGA;p.E604delinsDPSDNEYFYVDFREYEYDL KWE,	0,33		0,35
	WT1:NM_000378:exon6:c.1075_1088GTAGGG,	0,45		0,49
	WT1:NM_000378:exon6:c.1054_1055CAAGAG,	0,45		0,47
	IDH2:NM_002168:exon4:c.419G>A:p.R140Q,	0,51		0,51
AML-579	DNMT3A:NM_022552:exon23:c.2644C>T:p.R882C,	0,49	2,5%	0,48
	DNMT3A:NM_175629.2:exon23:c.2602T>C, p(.Phe868Leu)	0,51		0,48
	FLT3:NM_004119:exon14:c.1793_1794insCTACGTTGATTT CAGAGAATATGA;p.E598delinsDYVDFREYE,	0,99		0,98
	IDH1:NM_005896:exon4:c.395G>A:p.R132H,	0,49		0,54
	NPM1:NM_002520:exon11:c.859_860insTCTG;p.L287fs,	0,44		0,51

t-PDX = transgenic PDX; *TE = transduction efficiency; transduction with Luciferase vector (pCDH-EF-eFFly-T2A-mCherry; vector size approx. 8833 bp); # specific mutation not included in sequencing panel.

REVIEWER COMMENTS

Reviewer #3 (Remarks to the Author):

The authors have carefully considered all the points I raised in the first review. They have clarified key points in the text and undertaken additional experiments. IN particular, thank you for confirming the genetics of the PDX's were maintained. I also appreciate that given the COVID-19 situation that undertaking lengthy revisions would have been challenging and applaud the authors for removing data for AML-393 and AML-491, a decision I am sure they did not take lightly. The manuscript is much stronger and the data is DDIT4L is a welcome addition.

Reviewer #4 (Remarks to the Author):

The authors have carefully responded to all my concerns and comments.

Point by Point response to reviewer comments

Carlet et al., NCOMMS-20-24554-T

*In vivo inducible reverse genetics in patients' tumors
to identify individual therapeutic targets*

We are very grateful to all reviewers for reviewing and improving our manuscript and for accepting it for publication in *Nature Communications*.

REVIEWER COMMENTS

Reviewer #3 (Remarks to the Author):

The authors have carefully considered all the points I raised in the first review. They have clarified key points in the text and undertaken additional experiments. IN particular, thank you for confirming the genetics of the PDX's were maintained. I also appreciate that given the COVID-19 situation that undertaking lengthy revisions would have been challenging and applaud the authors for removing data for AML-393 and AML-491, a decision I am sure they did not take lightly. The manuscript is much stronger and the data is DDIT4L is a welcome addition.

We thank the reviewer for her/his careful evaluation of our revision and for the scientifically wise and encouraging comments.

Reviewer #4 (Remarks to the Author):

The authors have carefully responded to all my concerns and comments.

We thank the reviewer for appreciating our revised manuscript.